# Unveiling UV/IR mixing via symmetry defects:
# A view from topological entanglement entropy

**Jintae Kim**[1,2⋆], **Yun-Tak Oh**[3†], **Daniel Bulmash**[4,5‡] **and Jung Hoon Han**[1∘]

**1** Department of Physics, Sungkyunkwan University, Suwon 16419, Korea
**2** Institute of Basic Science, Sungkyunkwan University, Suwon 16419, South Korea
**3** Division of Display and Semiconductor Physics, Korea University, Sejong 30019, Korea
**4** Department of Physics, United States Naval Academy, Annapolis, MD 21402, USA
**5** Department of Physics and Center for Theory of Quantum Matter,
University of Colorado Boulder, Boulder, Colorado 80309, USA

⋆ jint1054@gmail.com , † ytak0105@gmail.com ,
‡ dbulmash@gmail.com , ∘ hanjemme@gmail.com

## Abstract

Some topological lattice models in two spatial dimensions exhibit intricate lattice size dependence in their ground state degeneracy (GSD). This and other features such as the position-dependent anyonic excitations are manifestations of UV/IR mixing. In the first part of this paper, we perform an exact calculation of the topological entanglement entropy (TEE) for a specific model, the rank-2 toric code. This analysis includes both contractible and non-contractible boundaries, with the minimum entropy states identified specifically for non-contractible boundaries. Our results show that TEE for a contractible boundary remains independent of lattice size, whereas TEE for non-contractible boundaries, similarly to the GSD, shows intricate lattice-size dependence. In the latter part of the paper we focus on the fact that the rank-2 toric code is an example of a translation symmetry-enriched topological phase, and show that viewing distinct lattice size as a consequence of different translation symmetry defects can explain both our TEE results and the GSD of the rank-2 toric code. Our work establishes the translation symmetry defect framework as a robust description of the UV/IR mixing in topological lattice models.

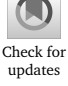

# 1 Introduction

Topological order in two dimensions are characterized by the emergence of anyonic excitations with fractional braiding statistics and the topology-dependent ground state degeneracy (GSD) [1, 2]. In the presence of symmetry, the classification of topological order becomes even richer as there can be multiple distinct "symmetry-enriched topological phases" (SETs) with the same anyons and statistics [3–8]. SETs exhibit symmetry fractionalization [3, 9], wherein the anyons transform projectively under the symmetry operations. Furthermore, the symmetry may non-trivially permute the anyon types, a phenomenon known as anyonic symmetry [8, 10, 11].

Typically, the GSD of a topologically ordered system on a torus depends only on the number of distinct anyon types and independent of the lattice size. However, several lattice models on a torus in which the GSD depends sensitively on the lattice size have been discussed recently [12–26]. For example, the $\mathbb{Z}_N$ plaquette model [12, 13] has GSD equal to $N^2$ for even×even lattice and $N$ for other cases, while other $\mathbb{Z}_N$ lattice models [17–26] on $L_x \times L_y$ torus have the GSD that depends on $L_x, L_y$ modulo $N$. This lattice size dependence has been viewed as a signature of UV/IR mixing [16, 27] in which the microscopic information of the theory such as the lattice size $L_x, L_y$ affects the low-energy, universal properties of the model such as the topological degeneracy of the ground state. For these models, anyons of different types arise depending on the positions at which they are located [19, 21, 26]. This follows in turn from the mobility constraints imposed on anyons in these models.

Another powerful explanation for the UV/IR mixing in the GSD is that these systems are translation SETs, which can be interpreted within the framework of the anyon condensation web in spatially modulated gauge theories [21, 22, 28, 29]. The translation SET nature of these models is evident, as different anyon types emerge at distinct positions, indicating that trans-

lation operations non-trivially permute the anyon types. Importantly, within the framework of translation SET, the lattice size dependence arises from translation symmetry defects threading non-contractible spatial cycles. This perspective has been employed to quantitatively explain the system size dependence in the GSD of certain lattice anyon model [26]. In the same paper, the topological entanglement entropy (TEE) for a contractible boundary was calculated and shown to be independent of the lattice size, which can also be understood from the symmetry defect picture. Another paper [25] computed the TEE for a contractible boundary in yet another anyon model with UV/IR mixing and found no system size dependence. It remains uncertain as to whether the UV/IR mixing is a special feature of GSD, or can be extended to the consideration of TEE as well.

In this work we compute the TEE for *non-contractible* boundaries [30–32] in a lattice anyon model with UV/IR mixing and show that lattice size dependence shows up in TEE for the non-contractible boundary, but not for the contractible boundary. We do this for the rank-2 toric code (R2TC) [17–21], known to show UV/IR mixing in the GSD. This is first shown through explicit calculations invoking the technique developed in [33]. Moreover, we show that this manifestation of lattice size dependence in TEE as well as GSD can be consistently understood in the framework of translation SET and translation symmetry defects. This is accomplished by matching formulas of TEE and GSD from explicit calculations with predictions based on theories of symmetry defects in SETs [10, 11, 34, 35]. In essence, for the $\mathbb{Z}_N \times \mathbb{Z}_N$ translation SET phase, any deviation in the linear size of the lattice from multiples of $N$ can be viewed as a symmetry defect and the powerful machinery developed in the past can be applied to sort out topological quantities in the presence of these defects. The same approach may be applied to other topological models with UV/IR mixing.

In Sec. 2, we provide an overview of the rank-2 toric code and introduce the position-dependent labels associated with anyonic excitations that are crucial for later understanding of translation symmetry defects. We derive the entanglement entropy of R2TC for both contractible and non-contractible boundaries in Sec. 3, and find in the latter case some lattice size dependence in the TEE. In Sec. 4, we show that both TEE and GSD of R2TC can be understood in the framework of translation SET and translation symmetry defects. Discussions are given in Sec. 5.

## 2 Rank-2 toric code

The $\mathbb{Z}_N$ R2TC [17–21], which can be obtained from the rank-2 lattice gauge theory through Higgsing [36, 37], is a stabilizer model on a square lattice with three mutually commuting stabilizers:

$$
\begin{aligned}
a_i &= Z_{0,i} Z_{0,i-\hat{x}}^{-1} Z_{0,i-\hat{y}}^{-1} Z_{0,i-\hat{x}-\hat{y}} Z_{2,i-\hat{y}} Z_{2,i}^{-2} Z_{2,i+\hat{y}} Z_{1,i-\hat{x}} Z_{1,i}^{-2} Z_{1,i+\hat{x}}, \\
b_i^x &= X_{2,i}^{1} X_{2,i+\hat{x}}^{-1} X_{0,i} X_{0,i-\hat{y}}^{-1}, \\
b_i^y &= X_{1,i} X_{1,i+\hat{y}}^{-1} X_{0,i} X_{0,i-\hat{x}}^{-1}.
\end{aligned}
\tag{1}
$$

At each site we have two $\mathbb{Z}_N$ degrees of freedom, denoted by subscripts 1 and 2, with an additional $\mathbb{Z}_N$ degree of freedom at the center of each plaquette denoted by the subscript 0. The $\mathbb{Z}_N$ Pauli operators $X$ and $Z$ form the algebra $Z_{i,a} X_{j,b} = \omega \delta_{ij} \delta_{ab} X_{j,b} Z_{i,a}$, where $a, b = 0, 1, 2$ and $\omega = e^{2\pi i/N}$. The "electric" stabilizer $a_i$ is centered at the site $i = (i_x, i_y)$; the two "magnetic" stabilizer $b_i^x, b_i^y$ are centered at the two links $(i, i+\hat{x})$ and $(i, i+\hat{y})$, respectively, as shown in Fig. 1.

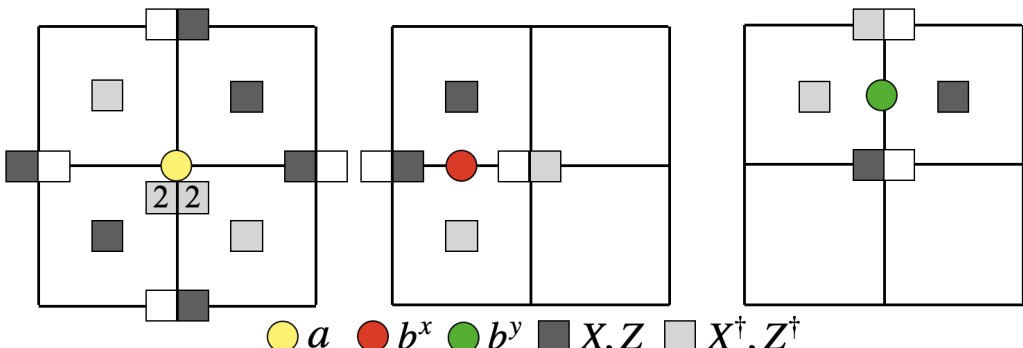

$\bigcirc a$  $\color{red}\bullet b^x$  $\color{green}\bullet b^y$  $\blacksquare X, Z$  $\square X^\dagger, Z^\dagger$

Figure 1: Three stabilizers $a_i$, $b_i^x$, and $b_i^y$ of R2TC are depicted as yellow, red, and green symbols, respectively. The $a_i$ stabilizer is a product of $\mathbb{Z}_N$ Pauli-$Z$ operators, while $b_i^x$ and $b_i^y$ consist of $\mathbb{Z}_N$ Pauli-$X$ operators. There are two $\mathbb{Z}_N$ spins defined at the vertices (represented by two adjacent squares) and one $\mathbb{Z}_N$ spin (represented by a single square) at the plaquette centers of the square lattice. The '2' inside the squares on the far left figure implies the action by $(Z^\dagger)^2$ for both spins. The empty square means there is no action by Pauli operators on that spin.

Excitations of this model are anyons carrying position-dependent labels [21]. We denote the anyon excitations associated with $a_i$, $b_i^x$, and $b_i^y$ as $[e]_i^{l_1}$, $[m^x]_i^{l_2}$, and $[m^y]_i^{l_3}$, respectively. For instance, $[e]_i^{l_1}$ is an eigenstate of $a_i$ with eigenvalue $\omega^{l_1}$. Similarly, $[m^x]_i^{l_2}$ ($[m^y]_i^{l_3}$) is an eigenstate of $b_i^x$ ($b_i^y$) with eigenvalue $\omega^{l_2}$ ($\omega^{l_3}$). The upper indices $l_1, l_2, l_3 \in \mathbb{Z}_N$ are the charges of anyons. Unlike the ordinary anyons, explicit coordinate dependence implies that anyons at different locations of the lattice correspond to different anyon types. For instance, $[e]_i^{l_1}$ and $[e]_j^{l_1}$ may represent different anyon types when $i \neq j$.

The coordinate dependence of the anyon types has been summarized in the formula [21]

$$
\begin{aligned}
[e]_i^{l_1} &= l_1 e + (l_1 i_x \bmod N) p^x + (l_1 i_y \bmod N) p^y \,, \\
[m^x]_i^{l_2} &= l_2 m^x + (l_2 i_y \bmod N) g \,, \\
[m^y]_i^{l_3} &= l_3 m^y - (l_3 i_x \bmod N) g \,.
\end{aligned}
\tag{2}
$$

Here $(e, p^x, p^y, m^x, m^y, g)$ are the six distinct Abelian anyon types, and the coefficients before each anyon type in Eq. (2) refer to how many anyons of a given type exist at that site.[1] Negative coefficients refer to anti-anyons. The electric anyon excitation $[e]_{0,0}^1$ corresponds to the anyon type $e$. When it is created at $i = (1, 0)$, it becomes $[e]_{1,0}^1 = e + p^x$, a sum of $e$ anyon type and $p^x$ anyon type. The $p^x$ and $p^y$ anyons are electric anyon dipoles consisting of ($[e]_{1,0}^1$, $[e]_{0,0}^{-1}$) pair and ($[e]_{0,1}^1$, $[e]_{0,0}^{-1}$) pair, respectively, as seen by directly calculating $[e]_{0,0}^{-1} + [e]_{1,0}^1 = p^x$ and $[e]_{0,0}^{-1} + [e]_{0,1}^1 = p^y$ using the first formula in Eq. (2). In a similar vein, $m^x$ and $m^y$ refer to the magnetic anyons $[m^x]_{0,0}^1$ and $[m^y]_{0,0}^1$, respectively. The $g$ anyon refers to a magnetic anyon dipole, which is a composite of either the ($[m^x]_{0,1}^1$, $[m^x]_{0,0}^{-1}$) pair or the ($[m^y]_{0,0}^1$, $[m^y]_{1,0}^{-1}$) pair [21]. The two definitions of the magnetic dipole are equivalent in the R2TC model. An important feature of the anyon contents in this model is that anyonic dipoles ought to be viewed as independent excitations rather than a mere composite of an anyon and an anti-anyon.

---

[1]More rigorously, the right-hand sides of Eq. 2 represent the fusion of anyons. In the case of $[e]_i^{l_1}$, it can be expressed as $[e]_i^{l_1} = \otimes_{n=1}^{l_1} e \otimes_{n=1}^{l_1 i_x \bmod N} p^x \otimes_{n=1}^{l_1 i_y \bmod N} p^y$

We can derive the following relations from Eq. (2):

$$
[e]^1_{i_x,i_y} = [e]^1_{i_x+N,i_y} = [e]^1_{i_x,i_y+N}\,,
$$
$$
[m^x]^1_{i_x,i_y} = [m^x]^1_{i_x+1,i_y} = [m^x]^1_{i_x,i_y+N}\,,
$$
$$
[m^y]^1_{i_x,i_y} = [m^y]^1_{i_x+N,i_y} = [m^y]^1_{i_x,i_y+1}\,. \tag{3}
$$

It means that for an $e$ anyon to preserve its anyon type, it must hop to a site $N$ lattice spacings away from the original one in either direction, resulting in *mobility restriction* on the $e$ anyon's motion. Similarly, an elementary $m^x$ ($m^y$) anyon can only hop by $N$ lattice spacing along $y$- ($x$-) direction but can freely move in the other [17, 19, 21]. Other motions at shorter lattice spacings are possible when the charge is greater than one. For example, $[e]^2_{i_x,i_y} = [e]^2_{i_x+2,i_y} = [e]^2_{i_x,i_y+2}$ for $\mathbb{Z}_4$ R2TC.

Using the R2TC model one can explicitly show that $(e,g)$, $(p^x,m^y)$, and $(p^y,-m^x)$ (minus sign means anti-anyon) pairs obey the same braiding statistics as the $(e,m)$ anyon pair in an ordinary $\mathbb{Z}_N$ toric code [17,18,21] and that one can view R2TC as three copies of $\mathbb{Z}_N$ regular $\mathbb{Z}_N$ toric codes. More precisely, the $\mathbb{Z}_N$ R2TC and the three copies of the $\mathbb{Z}_N$ toric code belong to the same topological phase *before* the symmetry considerations are made. The position-dependent anyon labels in Eq. (2) implies that R2TC is an SET with non-trivial anyonic symmetry under translation while three copies of $\mathbb{Z}_N$ toric codes have trivial anyonic symmetry under it.

The ground state(s) of R2TC is an eigenstate of all the stabilizers with the eigenvalue +1. The GSD of this model on a torus, first worked out for prime $N$ [17] and subsequently for arbitrary $N$ [21], is

$$
\text{GSD} = N^3 \gcd(L_x,N)\gcd(L_y,N)\gcd(L_x,L_y,N)\,, \tag{4}
$$

for $L_x \times L_y$ torus. We will discuss an efficient way of deriving the GSD formula using stabilizer identities in the following section.

## 2.1 Stabilizer identities in R2TC

The following three identities arise directly from the definition of stabilizers of R2TC:

$$
I_1 \equiv \prod_i a_i = 1\,, \quad I_2 \equiv \prod_i b^x_i = 1\,, \quad I_3 \equiv \prod_i b^y_i = 1\,. \tag{5}
$$

The product $\prod_i$ runs over all the sites of the torus. The identity arises because the same $X$ or $Z$ operator appears once as it is and once as its conjugate. They imply the conservation of the total $\mathbb{Z}_N$ electric ($I_1$) and two magnetic ($I_2$, $I_3$) charges. The other three identities are

$$
I_4 \equiv \left[\prod_i (a_i)^{i_x}\right]^{c_x} = 1\,, \qquad I_5 \equiv \left[\prod_i (a_i)^{i_y}\right]^{c_y} = 1\,, \qquad I_6 \equiv \left[\prod_i (b^x_i)^{-i_y}(b^y_i)^{i_x}\right]^{c_{xy}} = 1\,. \tag{6}
$$

Each stabilizer is raised to a power of the coordinate, and the identities $I_4$, $I_5$, $I_6$ refer to the conservation of electric dipoles in both $x$ and $y$ directions, and the combined of magnetic dipole moments we call the angular moment [17]. Importantly, the identities are obtained when the products are raised to appropriate "winding numbers" $c_x$, $c_y$, $c_{xy}$ given by

$$
c_x = N/\gcd(L_x,N) = \text{lcm}(L_x,N)/L_x\,,
$$
$$
c_y = N/\gcd(L_y,N) = \text{lcm}(L_y,N)/L_y\,,
$$
$$
c_{xy} = N/\gcd(L_x,L_y,N)\,. \tag{7}
$$

The necessity to introduce nontrivial values of $c_x, c_y, c_{xy} > 1$ arises from the fact that

$$\prod_i (a_i)^{i_x} = \prod_{i_y} Z^{L_x}_{1,\hat{x}+i_y\hat{y}} Z^{-L_x}_{1,L_x\hat{x}+i_y\hat{y}},$$

$$\prod_i (a_i)^{i_y} = \prod_{i_x} Z^{L_y}_{2,i_x\hat{x}+\hat{y}} Z^{-L_y}_{2,i_x\hat{x}+L_y\hat{y}},$$

$$\prod_i (b_i^x)^{-i_y}(b_i^y)^{i_x} = \prod_{i_y} X^{L_x}_{0,L_x\hat{x}+i_y\hat{y}} \prod_{i_x} X^{-L_y}_{0,i_x\hat{x}+L_y\hat{y}}, \tag{8}$$

are not equal to unity unless $L_x$, $L_y$ are multiples of $N$. To guarantee that the product becomes equal to unity for arbitrary lattice size, we need to find the smallest positive integers $c_x$, $c_y$, and $c_{xy}$ satisfying

$$L_x c_x \bmod N = 0, \quad L_y c_y \bmod N = 0,$$
$$L_x c_{xy} \bmod N = 0, \quad \& \quad L_y c_{xy} \bmod N = 0, \tag{9}$$

which are given in Eq. (7). We refer to the six identities $I_1$ through $I_6$ as the stabilizer identities of R2TC.

One can derive the GSD by comparing the number of stabilizer identities against the total number of stabilizers in the model. Each of the three identities $I_1, I_2, I_3$ generates $N$ constraints for the stabilizers since $(I_1)^{n_1} = (I_2)^{n_2} = (I_3)^{n_3} = 1$ for $1 \le n_1, n_2, n_3 \le N$. The GSD becomes $N \times N \times N = N^3$, but this is not all. From the three remaining identities we have $(I_\alpha)^{n_\alpha} = 1$ with $1 \le n_4 \le \gcd(L_x, N)$, $1 \le n_5 \le \gcd(L_y, N)$, and $1 \le n_6 \le \gcd(L_x, L_y, N)$, respectively. The full GSD formula of Eq. (4) is recovered in this manner.

Among the six identities, $\{I_2, I_3, I_6\}$ come from taking the product of magnetic stabilizers and $\{I_1, I_4, I_5\}$ from those of electric stabilizers. The fact that there are three identities for each type of stabilizers will play a crucial role in the evaluation of TEE.

## 2.2 Wegner-Wilson operators in the $X$-basis

There are six WW operators $W_1$ through $W_6$, given as the product of $X$-operators [19]:

$$W_{1,i_y} = \prod_{i_x=1}^{L_x} X_{0,i}, \qquad W_{2,i_x} = \prod_{i_y=1}^{L_y} X_{2,i},$$

$$W_{3,i_y} = \prod_{i_x=1}^{L_x} X_{1,i}, \qquad W_{4,i_x} = \prod_{i_y=1}^{L_y} X_{0,i},$$

$$W_{5,i_y} = \prod_{i_x=1}^{\mathrm{lcm}(L_x,N)} (X_{1,i})^{i_x}(X_{0,i})^{i_y},$$

$$W_{6,i_x} = \prod_{i_y=1}^{\mathrm{lcm}(L_y,N)} (X_{2,i})^{i_y}(X_{0,i})^{i_x}. \tag{10}$$

They represent the creation and annihilation process (CAP) of a $y$-oriented electric dipole and anti-dipole pair along the $x$- ($W_1$) and $y$-axis ($W_2$), of an $x$-oriented electric dipole and anti-dipole pair along the $x$- ($W_3$) and $y$-axis ($W_4$), and of an electric monopole and anti-monopole along the $x$- ($W_5$) and $y$-axis ($W_6$) as shown in Fig. 2 [19].

We note that $W_1$, $W_3$, and $W_5$ are defined along the $x$-axis, while $W_2$, $W_4$, and $W_6$ are defined along the $y$-axis. The $i_y$ of $W_1$, $W_3$, and $W_5$, as well as the $i_x$ of $W_2$, $W_4$, and $W_6$, can

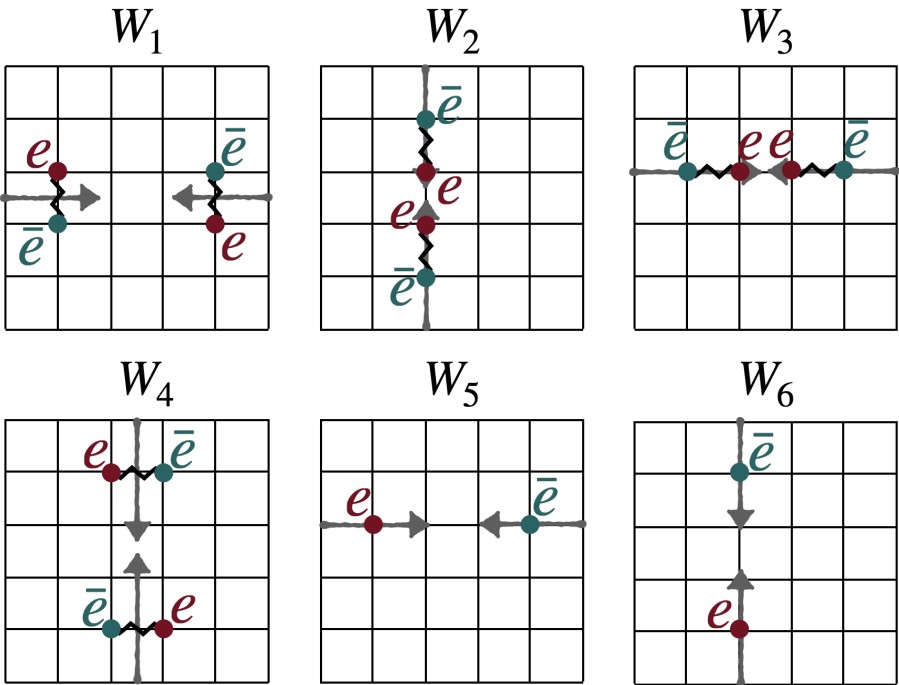

Figure 2: The CAPs associated with anyons and their corresponding anti-anyon pairs are illustrated for the respective WW operators.

be freely chosen due to their topological nature. In other words, two WW operators that only differ in position are not independent and can be connected by the product of $b_i^x$, $b_i^y$, and WW operators. Moreover, not all of WW operators are independent logical operators contributing to the GSD. In other words, certain WW operators may be expressed as the product of $b_i^x$, $b_i^y$, and other WW operators, contingent upon the lattice size.

The WW operators can also be used to evaluate the GSD [19]. To quote the final result, we have degeneracy factors of $N$ from $W_2$ and $W_3$ each, $\gcd(L_x, N)$ from $W_5$, $\gcd(L_y, N)$ from $W_6$, and the factor of $N \gcd(L_x, L_y, N)$ from $W_1$ and $W_4$ combined [19], for a total of $N^3 \gcd(L_x, N) \gcd(L_y, N) \gcd(L_x, L_y, N)$ independent logical operators. These WW operators are essential for constructing the minimum entropy states (MESs), which yields the maximum TEE for the region with non-contractible boundaries by appropriately combining the ground states.

## 3 Topological entanglement entropy of R2TC

We can calculate the TEE of the R2TC by adapting the method proposed in [25, 33], which is applicable when a state is expressed as

$$|\psi\rangle = |G|^{-1/2} \sum_{g \in G} g |0\rangle . \tag{11}$$

Here $|0\rangle$ is a product state satisfying $Z|0\rangle = |0\rangle$ for all the $Z$-operators. The group $G$ has elements that are products of $X$ operators. In the case of stabilizer models, the ground state(s) are generated by using $g$ made out of products of stabilizers and WW operators. (Note, however, that the stabilizers of the plaquette model are constructed as a product of both $X$ and $Z$ operators. This is why the method of [33] cannot be applied to compute the entanglement entropy of the plaquette model.)

Then, the reduced density matrix of the region $A$ for $|\psi\rangle$ obtained after dividing the lattice into two non-overlapping regions $A$ and $B$ is expressed as [33]:

$$\rho_A = |G|^{-1} \sum_{g \in G, g' \in G_A} g_A |0\rangle_A \langle 0|_A (g_A g'_A)^\dagger, \tag{12}$$

and the entanglement entropy (EE) of a region $A$ for the state $|\psi\rangle$ is [33]

$$S_A = \log\left(\frac{|G|}{|G_A||G_B|}\right). \tag{13}$$

We can separate each element $g$ as $g = g_A \otimes g_B$, where $g_A$ and $g_B$ act solely on the regions $A$ and $B$, respectively. Likewise, $|0\rangle = |0\rangle_A \otimes |0\rangle_B$. We introduce two sets, $G_A = \{g \in G | g = g_A \otimes I_B\}$, and $G_B = \{g \in G | g = I_A \otimes g_B\}$. Note that both Eqs. (12) and (13) remain valid for non-contractible as well as contractible boundaries. We review the derivation of Eq. (13) in Appendix A.

The appropriate group $G$ with which to construct the ground state of R2TC is generated by $b_i^x$, $b_i^y$ as well as a selection of WW operators from $W_1$ through $W_6$ given in Eq. (10). The exact choice of WW operators depends on the boundary we choose to calculate the TEE being contractible or not. Note that both stabilizers and WW operators are written in the $X$-basis and mutually commute. The group cardinality $|G|$ depends on how many WW operators we include besides the $b_i^x$, $b_i^y$'s.

## 3.1 Entanglement entropy: Contractible boundary

First, we consider the region $A$ in the form of $l_x \times l_y$ rectangle. The spins lying *on the boundary* are counted as living *inside A*. The ground state we use for the calculation of EE is $|\psi\rangle = |G|^{-1/2} \sum_{g \in G} g|0\rangle$, where $G$ is generated by two magnetic stabilizers $b_i^x, b_i^y$ but none of the WW operators. A naive estimate of $|G|$ for the group spanned by $(b_i^x, b_i^y)$ on a $L_x \times L_y$ lattice is $N^{2L_x L_y}$, ignoring the constraint among the stabilizers. However, there are three identities $I_2, I_3, I_6$ among the magnetic stabilizers that reduce the degrees of freedom by factors of $N, N, \gcd(L_x, L_y, N)$, respectively, and therefore

$$|G| = \frac{N^{2L_x L_y}}{N^2 \gcd(L_x, L_y, N)}. \tag{14}$$

We next compute $|G_A|$. The number of $b_i^x$'s and $b_i^y$'s acting entirely within the region $A$ is $N^{l_x(l_y-1)}$ and $N^{l_y(l_x-1)}$, respectively, and the cardinality of $G_A$ follows

$$
\begin{aligned}
|G_A| &= N^{l_x(l_y-1)} \times N^{l_y(l_x-1)} \\
&= N^{l_x(l_y-1)+l_y(l_x-1)},
\end{aligned} \tag{15}
$$

*before* considering three stabilizer identities $I_2 = I_3 = I_6 = 1$ in Eqs. (5) and (6). Because of these constraints, some stabilizers are no longer independent. On the other hand, we may as well choose those dependent stabilizers to lie entirely in the region $B$, then $|G_A|$ evaluated in Eq. (15) will be valid without loss of generality.

Now we come to the cardinality $|G_B|$. Since dependent stabilizer lie entirely in the region $B$, all $b_i^x$, $b_i^y$ stabilizers such that $b_i^x \notin G_B$ and $b_i^y \notin G_B$ are independent. Here $\notin$ means the stabilizer does not lie *entirely* within $B$, and include stabilizers defined exclusively in $A$ as well as those defined across both regions $A$ and $B$. Counting their numbers gives $(l_x+1)(l_y+2)$ and $(l_x+2)(l_y+1)$ for $b_i^x$ and $b_i^y$, respectively, leading to the naive count $|G'_B| = |G|/N^{(l_x+1)(l_y+2)+(l_x+2)(l_y+1)}$. In arrivng at this conclusion we used the fact that all the

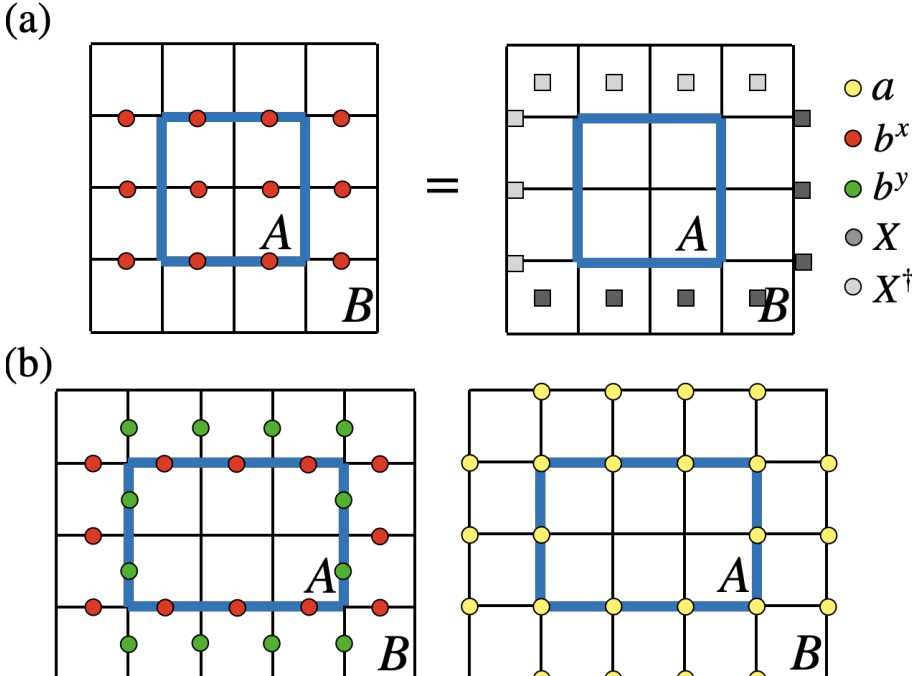

Figure 3: Region $A$ consists of spins on the blue boundary line and its interior. (a) Product of stabilizers in the region $A$ (left) is equal to the product of $X$ (and $X^\dagger$) operators along the boundary just outside of $A$ (right). (b) Marginal stabilizers among $b_i^x$, $b_i^y$'s (left) and $a_i$'s (right) having support in both $A$ and $B$ regions are indicated as colored circles.

dependent stabilizers are situated in $B$ by assumption, and therefore one can treat those stabilizers not entirely in $B$ (i.e. $b_i^x, b_i^y \notin G_B$) as independent and contributing to the cardinality count.

Note that in some cases an individual stabilizer does not lie fully inside $B$ but their product does, and qualifies as an element of $G_B$. For example, the products

$$\prod_{b_i^x \notin G_B} b_i^x, \qquad \prod_{b_i^y \notin G_B} b_i^y, \qquad \prod_{b_i^x, b_i^y \notin G_B} (b_i^x)^{-i_y}(b_i^y)^{i_x}, \tag{16}$$

span all the sites $i$ for which the stabilizers $b_i^x$ and $b_i^y$ do not lie fully in the region $B$. Despite the product covering a two-dimensional region, one can check that the $X$ operators in the interior of $A$ cancel out, leaving only the product of $X$'s and $X^\dagger$'s along the one-dimensional loop lying just outside the boundary of $A$ as illustrated in Fig. 3 (a). We refer to the three operators in Eq. (16) as *boundary operators*. Importantly, all the boundary operators consist of $X$ operators that are located in the region $B$, and are thus elements of $G_B$. Yet, the stabilizers that enter in the definition of boundary operators qualify the condition $b_i^x, b_i^y \notin G_B$ and have been "counted out" in the previous estimation of the cardinality of $G_B$. To correct for this mistake, we need to multiply $|G'_B|$ by $N^3$, as each boundary operator adds a factor $N$ to the cardinality:

$$\begin{aligned} |G_B| &= |G'_B| \times N^3 \\ &= |G| N^{3-(l_x+1)(l_y+2)-(l_y+1)(l_x+2)}. \end{aligned} \tag{17}$$

Combining the results of $|G|, |G_A|, |G_B|$ we arrive at the entanglement entropy of a disk-like bi-partition for R2TC:

$$
\begin{aligned}
S_A &= \log\left(\frac{|G|}{|G_A||G_B|}\right) \\
&= [4(l_x + l_y + 1) - 3]\log N .
\end{aligned}
\tag{18}
$$

The area law part of the entropy is proportional to $4(l_x + l_y + 1)$, equal to the number of *boundary stabilizers* that refer to a collection of $b_i^x$, $b_i^y$ stabilizers with support on both $A$ and $B$ as shown in Fig. 3 (b). The important remaining factor is $\gamma = 3\log N$, which we associate as the TEE of the R2TC. It shows no lattice size dependence, and a similar conclusion has been shown before using a different topological model. [25].

This value is also related to the fact that only three boundary operators can be constructed as in Eq. (16), regardless of the shape of the (contractible) boundary.

In hindsight, an independent calculation of the total cardinality $|G|$ as given in Eq. (14) was redundant, as it is only the ratio $|G|/|G_B|$ that is required for the calculation of EE. We will take advantage of this feature also in calculations of EE for the non-contractible boundary.

All ground states of R2TC share the same TEE as long as the boundary is contractible. To show this, we can derive the reduced density matrix of the region $A$ for general ground state of R2TC and show that it remains insensitive to the choice of ground states. The outline of the proof goes as follows. As stated earlier, we take $|\psi\rangle$ as the ground state of R2TC generated by the two stabilizers $b_i^x$, $b_i^y$ and none of the WW operators. Such a state has the eigenvalue $+1$ for all the logical operators written in the $Z$-basis. Other ground states of R2TC, which are also the eigenstates of logical operators in the $Z$-basis, can be constructed by $o_\alpha|\psi\rangle$, where $o_\alpha = \prod_{i=1}^6 (W_i)^{\alpha_i}$ with $1 \le \alpha_i \le N$ is the product of logical operators in the $X$-basis. There are at most $N^6$ such operators $o_\alpha$. Each $o_\alpha|\psi\rangle$ has different eigenvalues of $Z$-logical operators but it still remains a ground state. The most general ground state of R2TC can be written as a linear combination $|\psi^{\text{gen}}\rangle = \sum_\alpha c_\alpha o_\alpha |\psi\rangle$, with the coefficients $c_\alpha$'s satisfying $\sum_\alpha |c_\alpha|^2 = 1$. In Appendix B.1 we prove that the reduced density matrix $\rho_A^{\text{gen}} = \text{Tr}_B\left[|\psi^{\text{gen}}\rangle\langle\psi^{\text{gen}}|\right]$ is the same regardless of the coefficients $c_\alpha$. Thus, as the entanglement entropy depends only on the reduced matrix, both the EE and the TEE are same for all ground states.

## 3.2 Entanglement entropy: Non-contractible boundary

We now consider a pair of parallel and non-contractible boundaries along the $x$- or $y$-axis of the torus and call them $x$-cut and $y$-cut, respectively. For non-contractible boundaries, MES that are constructed as a simultaneous eigenstate of *all* logical operators running along the cut plays a significant role in the consideration of TEE [30, 38]. The choice of MES is not unique, but as we prove in Appendix B.2, all those MES yield the same TEE. Therefore, it suffices to construct the MES with eigenvalue $+1$ for all the logical operators running along the cut.

With that in mind, we construct the MES of R2TC for the $x$-cut as

$$
|\psi_{\text{MES}}\rangle = |G_{\text{MES}}|^{-1/2} \sum_{g \in G_{\text{MES}}} g |0\rangle ,
\tag{19}
$$

with the group $G_{\text{MES}}$ generated by $b_i^x$, $b_i^y$, $W_1$, $W_3$, and $W_5$. The other three WW operators $W_2, W_4, W_6$ run along the $y$-axis and do not qualify as logical operators in the case of the $x$-cut. The state in Eq. (19) is an eigenstate of $W_1, W_3, W_5$, as well as all logical operators in the $Z$-basis running along the $x$-axis with eigenvalue $+1$, and thus qualifies as MES regardless of the lattice size [30].

Consider the region $A$ of width $l_y$ bounded by two non-contractible boundaries running along the $x$-axis. In computing the cardinality we assume, without loss of generality, that the

dependent stabilizers are situated in the region $B$. Additionally, we assume that all logical operators lie entirely in the region $B$ and located as close as possible to the lower boundary of region $A$. Both assumptions are useful in simplifying the cardinality calculations.

We first compute $|G_B|$. Since by assumption all the dependent stabilizers and the logical operators are in the region $B$, we arrive at the cardinality of $G_B$ by dividing the total cardinality $|G_{\text{MES}}|$ by the the number of elements of $G_{\text{MES}}$ that do not belong to $B$. This is done, in turn, by counting the number of independent $b_i^x$ and $b_i^y$ stabilizers such that $b_i^x \notin G_B$ and $b_i^y \notin G_B$. By assumption, all such stabilizers are independent. There are $L_x(l_y + 2)$ and $L_x(l_y + 1)$ of them, respectively, and we get $|G'_B| = |G_{\text{MES}}|/N^{L_x(l_y+2)+L_x(l_y+1)}$. As in the case of contractible boundary discussed before, however, additional considerations have to be made for the following operators:

$$\prod_{b_i^x \notin G_B} b_i^x , \qquad \prod_{b_i^y \notin G_B} b_i^y , \qquad \left( \prod_{b_i^x, b_i^y \notin G_B} (b_i^x)^{-i_y}(b_i^y)^{i_x} \right)^{c_x} . \tag{20}$$

The first two expressions are identical to those of Eq. (16). The third expression is that of Eq. (16) raised to $c_x = N/\gcd(L_x, N)$ to take account of the periodic boundary conditions. They are made of stabilizers $\notin G_B$ and have been taken into consideration when calculating $|G'_B|$. However, as observed before, the product of stabilizers leaves a string of operators within the region $B$ and should count toward the elements of $G_B$. Each operator in Eq. (20) contributes a factor of $N, N$ and $\gcd(L_x, N)$ to $|G_B|$ and therefore the correct cardinality is

$$|G_B| = |G_{\text{MES}}| N^2 \gcd(L_x, N)/N^{L_x(l_y+2)+L_x(l_y+1)} . \tag{21}$$

Note that only the knowledge of the ratio $|G_{\text{MES}}|/|G_B|$, not $|G_{\text{MES}}|$ itself, is required for the calculation of entanglement entropy. No consideration is needed for the logical operators and their contributions to the cardinality as they are, by assumption, all in the region $B$.

Now we compute $|G_A|$. The number of independent $b_i^x$'s and $b_i^y$'s acting entirely in $A$ is $L_x(l_y - 1)$ and $L_x l_y$, respectively, and the naive count is $|G'_A| = N^{L_x(2l_y-1)}$. There are some additional operators in $G_A$ that can be expressed as the product of stabilizers not entirely in $A$. We have identified four such operators, with three of them being:

$$W_{1,q_y-1} \prod_{i_x=1}^{L_x} b_{i_x,q_y}^x , \qquad W_{3,q_y-1}^{-1} \prod_{i_x=1}^{L_x} b_{i_x,q_y-1}^y ,$$
$$W_{5,q_y-1}^{-1} \prod_{i_x=1}^{\text{lcm}(L_x,N)} (b_{i_x,q_y-1}^y)^{i_x} \prod_{i_x=1}^{L_x} (b_{i_x,q_y}^x)^{\frac{\text{lcm}(L_x,N)}{L_x}(q_y+1)} . \tag{22}$$

Here, $q_y$ is the $y$-coordinate of the lower boundary of the region $A$. The idea behind the construction is this: WW operators in the region $B$ (but lying close to the lower boundary of $A$) turn into WW operators in the region $A$ by being multiplied with a product of stabilizers at the border of $A$ and $B$ regions. The case for the first operator in Eq. (22) is depicted in Fig. 4.

A fourth operator is defined when $W_1^k$ with certain integer $k$ is not a logical operator. As mentioned earlier, whether $W_1^k$ serves as a logical operator or not depends on the lattice size relative to $N$. Suppose $W_1^k$ is *not* a logical operator after all, then it must be the case that $W_1^k$ becomes a product of $b_i^x$ and $b_i^y$ [19]. Keeping this in mind, the fourth operator in question is

$$W_{1,q_y-1}^k (W_{1,A})^\dagger \left( \prod_{i_x=1}^{L_x} b_{i_x,q_y}^x \right)^k \equiv W_{1,B} \left( \prod_{i_x=1}^{L_x} b_{i_x,q_y}^x \right)^k , \tag{23}$$

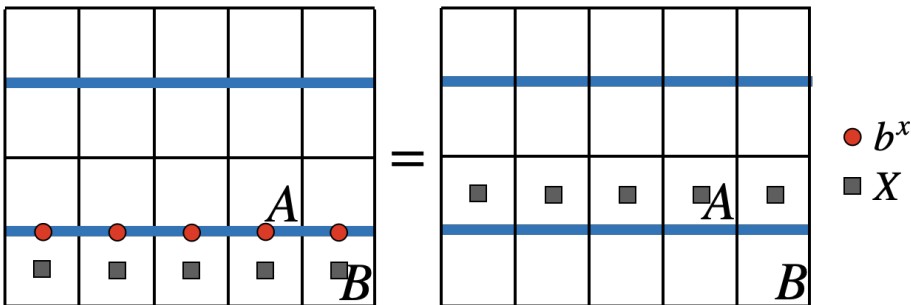

Figure 4: Illustration showing that $W_{1,q_y-1}\prod_{i_x=1}^{L_x}b_{i_x,q_y}^x$ is entirely within region $A$. The $y$-coordinate of the lower boundary of region $A$ is $q_y$.

where in general one can decompose $W_{1,q_y-1}^k = W_{1,A}W_{1,B}$ with $W_{1,A}$ being a product of $b_i^x, b_i^y \in G_A$ and $W_{1,B}$ a product of $b_i^x, b_i^y \notin G_A$. Since $b_{i_x,q_y}^x \neq G_A$, the above operator is made of stabilizers that are not entirely in $A$. Nevertheless the overall product consists of $X$-operators lying entirely in $A$. The operator in Eq. (23) is a product of three operators $W_{1,q_y-1}^k$, $\left(\prod_{i_x=1}^{L_x}b_{i_x,q_y}^x\right)^k$, and $W_{1,B}^\dagger$. The product of the first two, namely $(W_{1,q_y-1}\prod_{i_x=1}^{L_x}b_{i_x,q_y}^x)^k$, is simply the first expression in Eq. (22) raised to the power of $k$ and lies in the region $A$. Additionally, by definition, $(W_{1,A})^\dagger$ consists of stabilizers that are entirely in $A$. Therefore, Eq. (23) gives an operator defined entirely in region $A$.

Combining the first operator in Eq. (22) and the fourth operator constructed in Eq. (23), we have an operator constructed out of $W_1$ whether $W_1^k$ itself is a logical operator or not. Hence $W_1$ contributes a factor $N$ to the cardinality $|G_A|$. The $W_3$ in the second operator of Eq. (22) contributes an additional factor $N$ to the cardinality. Finally, $W_5$ appearing in the third operator in Eq. (22)) contributes $\gcd(L_x, N)$. To conclude, the cardinality of $G_A$ is $N^{L_x(2l_y-1)}N^2\gcd(L_x, N)$.

The results can be summarized:

$$|G_A| = N^{L_x(2l_y-1)}N^2\gcd(L_x, N),$$
$$|G_B| = |G_{\mathrm{MES}}|N^2\gcd(L_x, N)/N^{L_x(l_y+2)+L_x(l_y+1)}. \tag{24}$$

Therefore, the entanglement entropy of such MES can be obtained, $S_{\mathrm{MES}}^x = 4L_x\log N - \gamma_{\mathrm{MES}}^x$, with

$$\gamma_{\mathrm{MES}}^x = \log(N^4\gcd(L_x, N)^2). \tag{25}$$

A similar construction of MES for the $y$-cut uses $b_i^x$, $b_i^y$, $W_2$, $W_4$, and $W_6$ as generators and gives

$$\gamma_{\mathrm{MES}}^y = \log(N^4\gcd(L_y, N)^2). \tag{26}$$

We have successfully confirmed that the lattice size dependence not only exists in GSD [17, 21] [Eq. (4)] but also in TEE [Eqs. (25) and (26)] for the bi-partition by *non-contractible* loops. Having obtained explicit lattice size-dependent formulas, one may naturally suspect that some general principle underlies their behavior. Such principle, in terms of translation symmetry defect picture, is given in the next section.

## 4 Translation symmetry enriched topological phase

The $\mathbb{Z}_N$ R2TC Hamiltonian has translation symmetry in both $x$ and $y$ directions. However, acting on an one of its anyons with translation symmetry operator in general changes its type.

For example, an $e$ anyon of charge $l_1$ at the site $(0,0)$, expressed as $l_1 e$, transforms under translation $T_x$ to $l_1 e + l_1 p^x$ anyon on site $(1,0)$, and is therefore topologically distinct from the original anyon.

The charaterization of anyon types given in Eq. (2) is for a single electric or magnetic anyon. For a collection of all such anyons located across the lattice, we can employ a more general formula to express the overall anyon content in the system as

$$[l]_i = \sum_r \left( [e]_{i+r}^{l_{1,i+r}} + [m^x]_{i+r}^{l_{2,i+r}} + [m^y]_{i+r}^{l_{3,i+r}} \right), \tag{27}$$

where $\sum_r$ is the sum over all the lattice sites. Each expression in the sum, $[e]_{i+r}^{l_{1,i+r}}$, $[m^x]_{i+r}^{l_{2,i+r}}$ and $[m^y]_{i+r}^{l_{3,i+r}}$, follows the definition given in Eq. (2) with $(l_{1,i+r}, l_{2,i+r}, l_{3,i+r})$ referring to the $e, m^x, m^y$ anyon charges at the site $i + r$. Performing the sum over the lattice $\sum_r$ yields a simple formula for $[l]_i$,

$$\begin{aligned} [l]_i = &(l_1 \bmod N)e + (l_2 \bmod N)m^x + (l_3 \bmod N)m^y \\ &+ \left( (l_1 i_x + l_4) \bmod N \right)p^x + \left( (l_1 i_y + l_5) \bmod N \right)p^y \\ &+ \left( (l_2 i_y - l_3 i_x + l_6) \bmod N \right)g \,, \end{aligned} \tag{28}$$

where

$$l_1 = \sum_r l_{1,i+r} \,, \quad l_2 = \sum_r l_{2,i+r} \,, \quad l_3 = \sum_r l_{3,i+r} \,, \tag{29}$$

are the total anyon charges of $e$, $m^x$, and $m^y$ type. Explicit dependence on the coordinate $i$ drops out after the summation $\sum_r$. The other three integers

$$l_4 = \sum_r l_{1,r} r_x \,, \qquad l_5 = \sum_r l_{1,r} r_y \,, \qquad l_6 = \sum_r \left( l_{2,r} r_y - l_{3,r} r_x \right), \tag{30}$$

represent the total dipole moments. The equation (28) expresses the position-dependent anyon type of R2TC in the most general manner. Since the integers in Eq. (28) are defined mod $N$, there are $N^6$ distinct anyon types. It is also quite easy to understand which anyon types remain invariant under various translation operations $T_x^m T_y^n$ where $T_x, T_y$ refer to translation by one site in the $x, y$ direction. The knowledge of translation-invariant anyons will play a vital role when we interpret TEE and GSD from the perspective of translation symmetry defects.

We will interpret the R2TC as a $\mathbb{Z}_N \times \mathbb{Z}_N$ translation SET. The reasoning is as follows: From Eq. (3), both $T_x$ and $T_y$ generate anyon permutations of rank $N$, that is, $T_x^N$ and $T_y^N$ leave the anyon type unchanged. If we coarse-grain to an $N \times N$ unit cell, then we can think of $T_x^N = T_y^N = 1$ as manifestations of "on-site" symmetry group G $= \mathbb{Z}_N^{(x)} \times \mathbb{Z}_N^{(y)}$ (the superscript notation is just a reminder that $T_x$ generates the first copy of $\mathbb{Z}_N$ and $T_y$ the second). Although the symmetry is not actually on-site, we will treat it as such in what follows according to the crystalline equivalence principle [39]. Therefore, the R2TC should be viewed as a $\mathbb{Z}_N \times \mathbb{Z}_N$ translation SET. Then, we will consider the lattice size change under a symmetry defect picture.

It is well-understood how to characterize symmetry defects in the presence of anyons; the correct mathematical framework is a "G-crossed modular tensor category." [10,11,34,35] (To avoid confusion with the notation $G$ for the group of generators earlier, we use the roman G when referring to symmetry defects.) We briefly summarize the salient results of [10]. Given a G-enriched topological phase, there are many symmetry defects associated to a group element $\mathbf{g} \in$ G; we call the set of such defects $\mathcal{C}_{\mathbf{g}}$. We denote one such defect as $a_{\mathbf{g}}$, which is an element of $\mathcal{C}_{\mathbf{g}}$. Any one of the defects associated to $\mathbf{g}$ can be obtained from any other by fusing it with

some anyon. (If **0** is the identity element of G, then the set of **0** defects $C_0$ is just the set of anyons.)

The fusion and braiding of symmetry defects can be considered on a similar footing as those of anyons. The main difference between symmetry defects and anyons is that if one defect braids around another, it can change the defect type because it is acted upon by an element of the symmetry group. In particular, if $a_\mathbf{g}$ does a full braid with $b_\mathbf{h}$ ($\mathbf{h} \in$ G), then we say that $a_\mathbf{g}$ transforms as $a_\mathbf{g} \to {}^\mathbf{h} a_\mathbf{g}$. If it turns out that ${}^\mathbf{h} a_\mathbf{g} = a_\mathbf{g}$, that is, the defect type does not change under the action of $\mathbf{h}$, then we say that $a_\mathbf{g}$ is $\mathbf{h}$-invariant. We denote the set of $\mathbf{h}$-invariant $\mathbf{g}$ defects by $\mathcal{C}_\mathbf{g}^\mathbf{h}$. The set of $\mathbf{g}$-invariant anyons is denoted by $\mathcal{C}_0^\mathbf{g}$. The set of distinct $\mathbf{g}$ defect, previously denoted $\mathcal{C}_\mathbf{g}$, is equivalent to $\mathcal{C}_\mathbf{g}^\mathbf{0}$.

The cardinality of $\mathcal{C}_\mathbf{g}$ is

$$|\mathcal{C}_\mathbf{g}^\mathbf{0}| = |\mathcal{C}_0^\mathbf{g}| = \# \text{ of } \mathbf{g}-\text{invariant anyons.} \tag{31}$$

Here we used the fact that $|\mathcal{C}_\mathbf{g}^\mathbf{h}| = |\mathcal{C}_\mathbf{h}^\mathbf{g}|$ in general, and that $\mathcal{C}_0^\mathbf{g}$ by definition represents the set of $\mathbf{g}$-invariant anyons [10].

If all the anyons are Abelian, then all the symmetry defects belonging to the set $\mathcal{C}_\mathbf{g}$ have the same quantum dimension. Using the fact that the total quantum dimension of the $\mathbf{g}$ defects must equal the total quantum dimension of the anyons (denoted $\mathcal{D}$) [10], it follows that the quantum dimension $d_{a_\mathbf{g}}$ of any individual $\mathbf{g}$ defect is

$$d_{a_\mathbf{g}} = \sqrt{\frac{\# \text{ of anyons}}{\# \text{ of } \mathbf{g}-\text{invariant anyons}}}. \tag{32}$$

We now use the defect formalism to explain the TEE results of Sec. 3. For a cut with a contractible boundary, the TEE of a state with an anyon $a_0$ inside the region $A$ is [38, 40]

$$\gamma = \log \frac{\mathcal{D}}{d_{a_0}}. \tag{33}$$

In the ground state for $L_x$, $L_y$ divisible by $N$, there are no anyons inside $A$ (other than the vacuum itself) and $\gamma = \log \mathcal{D} = 3 \log N$ in the case of R2TC. An arbitrary $L_x, L_y$ means that defects $\mathbf{g}$ and $\mathbf{h}$ are piercing the non-contractible loops along $x$- and $y$-axis of the torus, respectively. Therefore, still only the vacuum is inside $A$, which means $\gamma = 3 \log N$, in agreement with our earlier result, Eq. (18). The defect picture gives a concise interpretation of the lack of lattice size dependence in the case of contractible boundary.

In the absence of symmetry defects, each MES for the $x$-cut can be associated in a one-to-one manner with a particular anyon $a_0 \in C_0$ crossing the $x$-cut cycle, and the TEE of such MES is given by [30, 38]

$$\gamma_{\text{MES}}^{x,(a_0)} = 2 \log \frac{\mathcal{D}}{d_{a_0}}. \tag{34}$$

For an Abelian model, $d_{a_0} = 1$ and $\mathcal{D}^2$ is the total number of anyons, so Eq. (34) becomes

$$\gamma_{\text{MES}}^{x,(a_0)} = \log (\# \text{ of anyons}). \tag{35}$$

For R2TC this equals $\gamma_{\text{MES}}^{x,(a_0)} = \log N^6$ in agreement with Eqs. (25) and (26).

Now, choosing a linear lattice size $L_x$ such that $N$ does not divide $L_x$ introduces a *CAP of symmetry defect* crossing the non-contractible loop along the $x$-axis of the torus. Therefore, defect comes in the form of non-contractible loop with no open ends. If we define

$$L_x \bmod N \equiv m, \quad L_y \bmod N \equiv n, \tag{36}$$

any excitation traversing the non-contractible loop along the $x$-axis of the torus will be acted upon by the element $\mathbf{g} = m$ of $\mathbb{Z}_N^{(x)}$ (in additive notation). Similarly, $L_y$ not divisible by $N$ gives rise to CAP of another symmetry defect $\mathbf{h} = n$ of $\mathbb{Z}_N^{(y)}$ crossing the non-contractible loop along the $y$-axis of the torus. Any excitation that traverses the non-contractible loop along the $y$-axis of the torus will be acted on by $\mathbf{h}$.

In this picture, each MES for the $x$-cut is associated with $a_{\mathbf{g}} \in C_{\mathbf{g}}^{\mathbf{h}}$. This is now $a_{\mathbf{g}}$ instead of $a_{\mathbf{0}}$, because, in effect, what is taking place for $m \neq 0$ is the CAP of a $\mathbf{g}$-symmetry defect and an anyon as a composite object rather than an anyon by itself. One can further argue that $a_{\mathbf{g}}$ is $\mathbf{h}$-invariant where $\mathbf{h}$ is the symmetry defect in the case of $L_y \mod N = n \neq 0$. Therefore, we conclude $a_{\mathbf{g}} \in C_{\mathbf{g}}^{\mathbf{h}}$.

If we could assume that the TEE formula analogous to Eq. (34) continues to hold in symmetry defect picture, the TEE of the MES associated with $a_{\mathbf{g}}$ is given by:

$$\gamma_{\text{MES}}^{x,(a_{\mathbf{g}})} = 2 \log \frac{\mathcal{D}}{d_{a_{\mathbf{g}}}} \,. \tag{37}$$

A similar mapping formulas valid for $C_{\mathbf{0}}$ (without symmetry defects) to an analogous formulas in the G-crossed modular tensor category (with symmetry defects) were used in an earlier work, such as GSD, modular $S$, and modular $T$ matrices [10]. According to Eqs. (32) and (37), for Abelian model,

$$\gamma_{\text{MES}}^{x,(a_{\mathbf{g}})} = \log\left(\# \text{ of } \mathbf{g}\text{–invariant anyons}\right). \tag{38}$$

We propose this as the formula capturing the TEE of an MES state associated with the symmetry defect $a_{\mathbf{g}}$. For the MES along the $x$-cut, the TEE is directly related to the number of $\mathbf{g}$-invariant anyons, where $\mathbf{g}$ is the symmetry defect pertaining to the size $L_x \mod N$. An analogous consideration gives

$$\gamma_{\text{MES}}^{y,(a_{\mathbf{h}})} = \log\left(\# \text{ of } \mathbf{h}\text{–invariant anyons}\right). \tag{39}$$

The two equations in Eqs. (38) and (39) capture the TEE of MES in the presence of translation symmetry defects. We can complete the argument by showing that these formulas indeed capture the TEE for non-contractible boundaries derived in the previous section.

In counting the number of $\mathbf{g}$- or $\mathbf{h}$-invariant anyons, we begin by considering how they transform as they move along the non-contractible loop of the torus and then return to their original position:

$$[l]_{L_x+i_x,i_y} = [l]_{i_x,i_y} + (l_1 L_x \mod N)p^x - (l_3 L_x \mod N)g \,. \tag{40}$$

For an anyon to be $\mathbf{g}$-invariant, it must satisfy $l_1 L_x \mod N = 0 = l_3 L_x \mod N$, that is, $l_1$ and $l_3$ must be multiples of $N/\gcd(L_x, N)$. Since $l_2$, $l_4$, $l_5$, and $l_6$ are in $\mathbb{Z}_N$ without restriction, we find the number of $\mathbf{g}$-invariant anyons to be $|C_{\mathbf{0}}^{\mathbf{g}}| = N^4 \gcd(L_x, N)^2$. Plugging this result into Eq. (38), we find perfect agreement with $\gamma_{\text{MES}}^x$ in Eq. (25). A similar reasoning correctly captures $\gamma_{\text{MES}}^y$ in Eq. (26). To conclude, the TEE formulas (38) and (39) obtained from the symmetry defect picture, coupled with the explicit anyon data such as Eq. (28), allows one to obtain the TEE of MES without going through arduous calculations such as performed in the previous section.

The utility of the symmetry defect picture goes beyond that of calculating TEE. It turns out that even the GSD formula of Eq. (4) can be recovered in the same language. G-crossed modularity [10] requires that

$$\begin{aligned}
\text{GSD}(\mathbf{g}, \mathbf{h}) &= \# \text{ of } \mathbf{g}\text{–invariant } \mathbf{h} \text{ defects}\\
&= \# \text{ of } \mathbf{h}\text{–invariant } \mathbf{g} \text{ defects.}
\end{aligned} \tag{41}$$

That is, we can calculate the GSD of R2TC using Eq. (41) by appropriately counting the number of invariant anyons. This is done by first identifying distinct defects in $C_{\mathbf{g}}$, then further identifying a subset of those that are invariant under the symmetry defect $\mathbf{h}$.

We can express general defects in $C_{\mathbf{g}}$ as $a_{\mathbf{g}} \otimes [l]_i$, where $a_{\mathbf{g}} \in C_{\mathbf{g}}$ is $\mathbf{h}$-invariant. First of all, any one of the defects in $C_{\mathbf{g}}$ can be obtained from any other by fusing it with some anyon. Furthermore, we can find at least one defect $a_{\mathbf{g}} \in C_{\mathbf{g}}$ which is $\mathbf{h}$-invariant, since the coexistence of both CAP of $\mathbf{g}$ and $\mathbf{h}$ defects implies that an $\mathbf{h}$-invariant defect must exist among $C_{\mathbf{g}}$. In the case of translation symmetry defects, allowing for arbitrary lattice size $L_x \times L_y$ is equivalent to the coexistence of $\mathbf{g}$ and $\mathbf{h}$ defects.

Not all of the defects in $a_{\mathbf{g}} \otimes [l]_i$ are distinct; we need to find a minimal set of anyons out of $[l]_i$ that, when fused with $a_{\mathbf{g}}$, generates all distinct defects in $C_{\mathbf{g}}$. To this end, we employ the known property that all defects in $C_{\mathbf{g}}$ are automatically $\mathbf{g}$-invariant [10]. The statement of $\mathbf{g}$-invariance becomes

$$a_{\mathbf{g}} \otimes [l]_{i_x,i_y} = {}^{\mathbf{g}}\left(a_{\mathbf{g}} \otimes l_{i_x,i_y}\right) = a_{\mathbf{g}} \otimes [l]_{L_x+i_x,i_y} \, . \tag{42}$$

Using Eq. (28), we can equivalently state Eq. (42) as

$$a_{\mathbf{g}} \otimes (l_1 L_x \bmod N) p^x \otimes (-l_3 L_x \bmod N) g = a_{\mathbf{g}} \, . \tag{43}$$

That is, in order for $a_{\mathbf{g}} \otimes l_i$ to be $\mathbf{g}$-invariant, $a_{\mathbf{g}}$ must reproduce itself when fused with $(l_1 L_x \bmod N) p^x$ or $(-l_3 L_x \bmod N) g$ for arbitrary $l_1, l_3$. By Bezout's identity, $l_1 L_x \bmod N, l_3 L_x \bmod N \geq \gcd(L_x, N)$. The distinct defects in $C_{\mathbf{g}}$, denoted as $a_{\mathbf{g}} \otimes [l]_i^{\mathbf{g}}$, are given by the fusion of arbitrary $\mathbf{h}$-invariant $\mathbf{g}$-defect $a_{\mathbf{g}}$ and

$$[l]_i^{\mathbf{g}} = l_1 e + l_2 m^x + l_3 m^y + ((l_1 i_x + l_4) \bmod \gcd(L_x, N)) p^x$$
$$+ ((l_1 i_y + l_5) \bmod N) p^y + ((l_2 i_y - l_3 i_x + l_6) \bmod \gcd(L_x, N)) g \, . \tag{44}$$

The difference between $[l]_i^{\mathbf{g}}$ and the general expression $[l]_i$ given in Eq. (28) is that the coefficients before $p^x$ and $g$ are both mod $\gcd(L_x, N)$ rather than mod $N$. The cardinality of $C_{\mathbf{g}}$ is $N^4 \gcd(L_x, N)^2$. The subset of $C_{\mathbf{g}}$ that are $\mathbf{h}$-invariant satisfies the additional requirement

$$[l]_{i_x,i_y}^{\mathbf{g}} = [l]_{i_x,L_y+i_y}^{\mathbf{g}} \, ,$$

which is fulfilled if

$$l_1 L_y \bmod N = 0 = l_2 L_y \bmod \gcd(L_x, N) \, . \tag{45}$$

This implies that $l_1$ and $l_2$ must be multiples of $N/\gcd(L_y, N)$ and $\gcd(L_x, N)/\gcd(L_x, L_y, N)$, respectively. Taking this addition constraint into account gives

$$|C_{\mathbf{g}}^{\mathbf{h}}| = |C_{\mathbf{g}}| \cdot \frac{\gcd(L_y, N)}{N} \cdot \frac{\gcd(L_x, L_y, N)}{\gcd(L_x, N)}$$
$$= N^3 \gcd(L_x, N) \gcd(L_y, N) \gcd(L_x, L_y, N) \, ,$$

equal to GSD in Eq. (4). Similar consideration applies to $|C_{\mathbf{h}}^{\mathbf{g}}|$ with the same result.

We conclude that various lattice size dependence manifested in the TEE and GSD of topological lattice models can be adequately explained by thinking of the system as a coarse-grained system enriched by a $\mathbb{Z}_N \times \mathbb{Z}_N$ symmetry which arises from lattice translations. Though our calculations are confined to one specific model, namely R2TC, we speculate that other models demonstrating lattice size dependence in various topological quantities may also fall within the same symmetry defect picture, with the translation symmetry playing the role of enriching the symmetry to SET.

## 5   Discussion

In conclusion, we have performed systematic calculations of the TEE for the rank-2 toric code, revealing that TEE for a contractible boundary remains unaffected by lattice size. Conversely, the TEE for non-contractible boundaries exhibits lattice-size dependence, similar to that of the ground state degeneracy in the same model. We quantitatively explain both results by viewing the R2TC as a translation SET with the lattice size itself acting as symmetry defects. Our work shows that the symmetry defect picture is powerful enough to predict topological properties such as the ground state degeneracy and the topological entanglement entropy.

Given the wide range of unique features in such translation SETs, it would be highly interesting to realize such models in spin, Rydberg atom, or other synthetic systems. It may also be possible to use our understanding of such translation SETs in (2+1) dimensions as a stepping stone to a more general understanding of (3+1)-dimensional phases with lattice size-dependent GSDs, including translation SETs and fracton topological orders.

Another potential avenue for future research is the exploration of the rotation+SWAP SET aspect of R2TC and its associated symmetry defect. By utilizing Eq. (2), it can be demonstrated that a $\pi/2$ rotation about the reference point $(i_x, i_y) = (0,0)$, combined with the exchange of qudits at the same vertices—achievable through the application of SWAP gates—induces the following transformations:

$$
\begin{aligned}
[e]_{i_x,i_y}^{l_1} &\to [e]_{-i_y,i_x}^{l_1}\,, \\
[m^x]_{i_x,i_y}^{l_2} &\to [m^y]_{-i_y,i_x}^{l_2}\,, \\
[m^y]_{i_x,i_y}^{l_3} &\to [m^x]_{-i_y,i_x}^{l_3}\,,
\end{aligned}
\tag{46}
$$

which are the permutation of anyon types. Morevoer, one may investigate the CAP of symmetry defects that induce the transformation in Eq. (46) and analyze its impact on TEE.

A few additional remarks are in order. One can use the exact tensor network ground state wave function for R2TC proposed in [19] to compute the entanglement entropy numerically. Without adopting the numerical scheme such as the tensor-netork renormalization group, it turns out the calculation is currently limited to $N = 2$ and $L_y \le 3$ for the case of the $y$-cut, with $L_x \le 5$, due to severe computational costs. Following the procedure outlined in [41], we numerically find the entanglement entropy in excellent agreement with $S_{\text{MES}}^y = 4L_y \log N - \log(N^4 \gcd(L_y, N)^2)$ for $N = 2$, $L_y = 3$ and $L_x = 4, 5$, confirming the validity of our derivation. Secondly, it is known that spurious TEE can appear for some non-topological states [26, 42, 43]. To check that our results are not being plagued by such effects, we use the model proposed in [43] and construct bulk products similar to Eq. (16). After the interior terms again cancel out, leaving the product of operators along the boundary of $A$. Importantly, this product does not completely encircle the boundary, as is the case in R2TC and other topological models, but is limited to a finite segment. Consequently, the TEE depends on the shape of the boundary and the area of the region (see Fig. 5). With this observation, we can ensure that the TEE of the R2TC is a genuine TEE. This comprehension can serve as a useful diagnostic for distinguishing between spurious and genuine TEE.

## Acknowledgments

We are grateful to Isaac Kim for helpful discussions and feedback, and to Yizhi You for an earlier collaboration.

**Funding information**  Y.-T. O. acknowledges support from the National Research Foundation of Korea (NRF), funded by the Korean government (MSIT), under grants No. RS-2023-00220471 and NRF-2022R1I1A1A01065149. D.B. was supported for part of this work by the Simons Collaboration on Ultra-Quantum Matter, which is a grant from the Simons Foundation (No. 651440). J.H.H. was supported by the National Research Foundation of Korea(NRF) grant funded by the Korea government(MSIT) (No. 2023R1A2C1002644). He acknowledges KITP, supported in part by the National Science Foundation under Grant No. NSF PHY-1748958, where this work was finalized.

## A  Entanglement entropy formula

In this section, we summarize a derivation of the entanglement entropy expression presented in Eq. (13) [25, 33]. The entanglement entropy $S_A$ of the region $A$ is formally defined as

$$S_A = \lim_{n \to 1} \frac{1}{1-n} \log \mathrm{Tr}[\rho_A^n], \tag{A.1}$$

where $\rho_A = \mathrm{Tr}_B[\rho]$ is the reduced density matrix of region $A$. Starting from the pure state expression in Eq. (11), $\rho_A$ can be expressed as

$$
\begin{aligned}
\rho_A &= \mathrm{Tr}_B[|\psi\rangle \langle\psi|] \\
&= |G|^{-1} \mathrm{Tr}_B\Big[ \sum_{g \in G, g' \in G} g\,|0\rangle \langle 0|\,(gg')^\dagger \Big] \\
&= |G|^{-1} \mathrm{Tr}_B\Big[ \sum_{g \in G, g' \in G} g_A\,|0\rangle_A \langle 0|_A\,(g_A g'_A)^\dagger \otimes g_B\,|0\rangle_B \langle 0|_B\,(g_B g'_B)^\dagger \Big] \\
&= |G|^{-1} \sum_{g \in G, g' \in G} g_A\,|0\rangle_A \langle 0|_A\,(g_A g'_A)^\dagger \times \langle 0|_B\,(g_B g'_B)^\dagger g_B\,|0\rangle_B \,,
\end{aligned}
\tag{A.2}
$$

where $g = g_A \otimes g_B$ and $|0\rangle = |0\rangle_A \otimes |0\rangle_B$. The requisite condition for the non-vanishing of $\langle 0|_B\,(g_B g'_B)^\dagger g_B\,|0\rangle_B$ is $g'_B = I_B$. Subsequently, the formulation for $\rho_A$ adopts the structure

$$\rho_A = |G|^{-1} \sum_{g \in G, g' \in G_A} g_A\,|0\rangle_A \langle 0|A(g_A g'_A)^\dagger, \tag{A.3}$$

where $G_A = \{g \in G | g = g_A \otimes I_B\}$. Employing Eq. (A.3), we proceed to deduce $\rho_A^2 = |G_A||G_B|/|G|\rho_A$ [25, 33]. After all, we arrive at

$$S_A = \lim_{n \to 1} \frac{1}{1-n} \log \mathrm{Tr}\left[\left(\frac{|G_A||G_B|}{|G|}\right)^{n-1} \rho_A\right] = \log\left(\frac{|G|}{|G_A||G_B|}\right) \tag{A.4}$$

as the conclusive expression.

## B  Reduced density matrix equivalence

In this section, we will prove that i) the reduced density matrix of the contractible boundary is the same for any arbitrary ground state of the R2TC, and ii) the EE and TEE for the $x$-cut is the same for any arbitrary MES of the R2TC.

## B.1 Contractible boundary

For the region $A$ in the form of rectangle surrounded by a contractible boundary, an important caveat in defining the logical operators is that we can always choose them in such a way that they are defined entirely in the region $B$, without crossing the region $A$ surrounded by a contractible boundary.

Then, the most general ground state of R2TC can be written as $|\psi^{\text{gen}}\rangle = \sum_\alpha c_\alpha o_\alpha |\psi\rangle$, with the coefficients $c_\alpha$ satisfying $\sum_\alpha |c_\alpha|^2 = 1$, and all $o_\alpha$ are entirely located in the region $B$. The reduced density matrix of $|\psi^{\text{gen}}\rangle$, denoted as $\rho_A^{\text{gen}}$, can be expressed as:

$$
\begin{aligned}
\rho_A^{\text{gen}} = \text{Tr}_B\left[|\psi^{\text{gen}}\rangle\langle\psi^{\text{gen}}|\right] &= |G|^{-1}\text{Tr}_B\left[\sum_{\alpha,\beta}\sum_{g\in G, g'\in G} c_\alpha c_\beta^* o_\alpha g |0\rangle\langle 0| (gg')^\dagger o_\beta^\dagger\right] \\
&= |G|^{-1}\text{Tr}_B\left[\sum_{\alpha,\beta}\sum_{g\in G, g'\in G} c_\alpha c_\beta^* g_A |0\rangle_A \langle 0|_A (g_A g_A')^\dagger \otimes o_\alpha g_B |0\rangle_B \langle 0|_B (g_B g_B')^\dagger o_\beta^\dagger\right] \\
&= |G|^{-1}\sum_{\alpha,\beta}\sum_{g\in G, g'\in G} c_\alpha c_\beta^* \left(\langle 0|_B (g_B g_B')^\dagger o_\beta^\dagger o_\alpha g_B |0\rangle_B\right) g_A |0\rangle_A \langle 0|_A (g_A g_A')^\dagger.
\end{aligned}
\tag{B.1}
$$

The matrix element $\langle 0|_B (g_B g_B')^\dagger o_\beta^\dagger o_\alpha g_B |0\rangle_B$ is nonzero only if the product of operators equals one. This arises from the fact that each of the operators in $(g_B g_B')^\dagger o_\beta^\dagger o_\alpha g_B$ is a product of $X$'s, hence the only way to obtain nonzero elements for $|0\rangle_B$ is for their product to become one. In other words,

$$
o_\beta^\dagger o_\alpha = g_B g_B' g_B^\dagger.
\tag{B.2}
$$

The expression on the right is a product of stabilizers acting on region $B$ and cannot equal the product of logical operators on the left, unless $o_\alpha = o_\beta$. It then follows $g_B' = I_B$. Ultimately, the expression for $\rho_A^{\text{gen}}$ can be simplified to:

$$
\rho_A^{\text{gen}} = |G|^{-1}\sum_{g\in G, g'\in G_A} g_A |0\rangle_A \langle 0|_A (g_A g_A')^\dagger,
\tag{B.3}
$$

where $G_A = \{g\in G | g = g_A \otimes I_B\}$. Here, $g_A$ and $g_A'$ are parts of $g$ and $g'$ acting on region $A$, respectively, and the number of $g_A$ and $g_A'$ may differ in general. One can readily see that $\rho_A^{\text{gen}}$ is equal to $\rho_A$ in Eq. (12).

## B.2 Non-contractible boundaries

When the MES with eigenvalue $+1$ for all the logical operators running along the $x$-cut is $|\psi_{\text{MES}}\rangle$, the general MES can be described by $U|\psi_{\text{MES}}\rangle$, where $U$ is the product of logical operators along the $y$ ($x$)-axis in the case of $x$ ($y$)-cut. Given that $U = U_A \otimes U_B$, the reduced density matrix of the region $A$ for a general MES, denoted as $\rho_A^{\text{gMES}}$, is expressed as follows:

$$
\begin{aligned}
\rho_A^{\text{gMES}} &= \text{Tr}_B[U|\psi_{\text{MES}}\rangle\langle\psi_{\text{MES}}|U^\dagger] \\
&= |G_{\text{MES}}|^{-1}\text{Tr}_B[\sum_{g,g'\in G} U_A g_A |0\rangle_A \langle 0|_A (g_A g_A')^\dagger U_A^\dagger \otimes U_B g_B |0\rangle_B \langle 0|_B (g_B g_B')^\dagger U_B^\dagger] \\
&= |G_{\text{MES}}|^{-1}\sum_{g\in G, g'\in G_A} U_A g_A |0\rangle_A \langle 0|_A (g_A g_A')^\dagger U_A^\dagger = U_A \rho_A^{\text{MES}} U_A^\dagger,
\end{aligned}
\tag{B.4}
$$

where $U = U_A \otimes U_B$. $\rho_A^{\text{MES}}$ equals $\rho_A$ in Eq. (12) since $|\psi_{\text{MES}}\rangle$ can be expressed as Eq. (11). Then, since $\text{Tr}_A[(\rho_A^{\text{gMES}})^n] = \text{Tr}_A[(\rho_A^{\text{MES}})^n]$, the EE and TEE outcomes for all MES are same.

## C  Spurious entanglement entropy

We analyze the spurious entanglement of the non-topological model proposed in [43], using our idea of taking the bulk product of stabilizers to arrive at the boundary product. We conclude the boundary product in this case does not fully encircle the region $A$, and also find some dependence on the shape of the boundary.

The model [43] is defined on a square lattice, comprising two stabilizers:

$$
a = \begin{matrix} IZ - IZ \\ | \quad | \\ IZ - XZ \end{matrix} \, , \quad b = \begin{matrix} ZX - ZI \\ | \quad | \\ ZI - ZI \end{matrix} \, ,
$$

(C.1)

Two $\mathbb{Z}_2$ degrees of freedom are assigned to each vertex corresponding to the first and the second Pauli operator in the above. The (unique) ground state of this model is

$$
|\psi\rangle = |G|^{-1/2} \sum_{g \in G} g \, |0+\rangle \, ,
$$

(C.2)

where the group $G$ is generated by $a$'s and $|0+\rangle$ satisfies $IX |0+\rangle = ZI |0+\rangle = |0+\rangle$ for arbitrary vertices.

We evaluate the entanglement entropy of $|\psi\rangle$ first for a $l_x \times l_y$ square region $A$. The operations that significantly influence the TEE are the elements of $G_B$, which are the product of $a \notin G_B$, and the elements of $G_A$, which are the product of $a \notin G_A$. We can identify the operation satisfying the former condition, depicted in Fig. 5 (a). The operation is the multiplication of $a$'s and encircled by the yellow loop. This operation in terms of $X$ and $Z$ takes the form of a square bracket encompassing a part of the boundary, signifying that the count of such operations is expected to vary based on the shape of the region. The entanglement entropy result is summarized as

$$
\begin{aligned}
|G| &= 2^{L_x L_y} \, , \\
|G_A| &= 2^{l_x l_y} \, , \\
|G_B| &= |G| 2^{-(l_x+2)(l_y+2)+1} \, , \\
S_A &= (2l_x + 2l_y) \log 2 - \log 2 \, ,
\end{aligned}
$$

(C.3)

and TEE is equal to $\log 2$. We can easily check that the TEE depends on the shape of the boundary. By altering the boundary, as depicted in Fig. 5 (b), we can identify two operations influential to the TEE, each one encircled by yellow loop.

This observation deviates from the well-known concept of TEE and should be regarded as spurious TEE. Conversely, if the operation takes the form of a WW loop encircling the entire boundary of the region, the number of elements cannot depend on the shape of the region, indicating a genuine TEE. Based on the analysis conducted in this simple example, we can confidently assert that there are no instances of spurious TEE in the entanglement entropy calculation for R2TC. The identified operations are the the form of WW loop, thereby eliminating any possibility of spurious TEE contributions.

## D  Tensor network calculation: Entanglement entropy for R2TC

In this section, we demonstrate how to express the MES of R2TC using a Tensor Network (TN) framework, building upon the TN wavefunction introduced in Ref. [19]. By examining the local gauge symmetry of tensors as summarized in Ref. [19], when both lattice sizes $L_x$

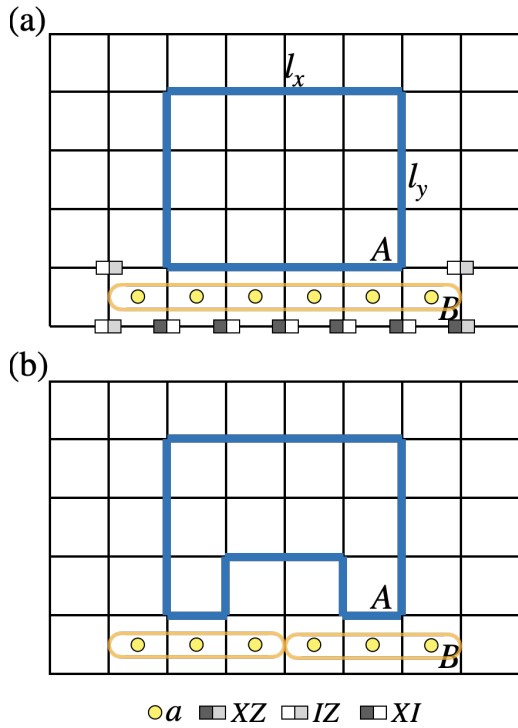

Figure 5: (a) An operation satisfying the condition of being an element of $G_B$ and the product of $a \notin G_A$. The operation is influential to the TEE and represented by by the encirculed yellow loop. (b) Two operations, represented by yellow loops and influential to the TEE, are depicted.

and $L_y$ are multiples of $N$, the TN wavefunction, or projected entangled-pair states (PEPS) wavefunction, serves as a ground state of R2TC and is simultaneously an eigenstate of $W_1$, $W_2$, $W_3$ and $W_4$, all with eigenvalues equal to 1. The WW operators $W_5$ and $W_6$ transition the PEPS ground state wavefunction to the other ground states without affecting the eigenvalues of $W_1$, $W_2$, $W_3$, and $W_4$.

With a $y$-directed entanglement cut, the MES should simultaneously be an eigenstate of the WW operators $W_2$, $W_4$, and $W_6$. Meanwhile, $W_1$, $W_3$, and $W_5$ should facilitate the transition of a given MES to other anyon sectors. From this perspective, the PEPS wavefunction initially proposed in Ref. [19] is not an MES and requires modifications to transform it into one.

The process of modifying the PEPS wavefunction to become an MES for a $y$-cut can be divided into two distinct steps. First, the PEPS wavefunction is projected so that it no longer serves as an eigenstate of $W_1$ and $W_3$. The projected ground state remains a ground state of the R2TC, with $W_1$ and $W_3$ now transitioning the projected ground state to another orthogonal ground state. In the second step, the wavefunction is further constrained to be an eigenstate of $W_6$, thereby completing the MES PEPS representation.

## D.1 Review: PEPS wavefunction

In this subsection, we review the PEPS wavefunction introduced in Ref. [19], which is suggested as one of the PEPS formulas responsible for one of the R2TC ground states. The PEPS

is composed of three local tensors $T_{udlr}$, $g^{s_1 s_2}_{udlr}$, and $g^{s_0}_{udlr}$ and can be represented as below:

$$ \tag{D.1} $$

The indices $s_1, s_2, s_0$ represent physical indices corresponding to the local $\mathbb{Z}_N$ orbitals of R2TC, while the indices $u, d, l, r$ are virtual indices with $\mathbb{Z}_N$ character that are contracted out in the finial expression. The first two tensors are defined as

$$ T_{udlr} = \delta_{u+r-(l+d),0}, \qquad g^{s_1 s_2}_{udlr} = \left(\delta_{l,r}\delta_{l,s_1}\right)\left(\delta_{d,u}\delta_{d,s_2}\right). \tag{D.2} $$

Note that the delta is implemented by mod $N$ throughout this section. To define the third local tensor, we need to introduce following isometry tensor:

$$ P^{s_0}_{s_1 s_1} = \delta_{s_0, s_1 + s_2}. \tag{D.3} $$

Then the third local tensor is defined as

$$ g^{s_0}_{udlr} = P^{s_0}_{s_1 s_1} g^{s_1 s_2}_{udlr}. \tag{D.4} $$

Here, the summation for repeated indices is implied.

## D.2 Loop-gas interpretation

As mentioned in above, the PEPS wavefunction is an eigenstate of WW operators, $W_1$, $W_2$, $W_3$, and $W_4$. To gain an intuitive understanding of this, we introduce the concept of a loop-gas configuration picture. From this point onward, we confine our analysis to the $\mathbb{Z}_2$ case for the sake of clarity and simplicity in the subsequent discussion. The concepts and principles discussed can be readily extended to the general $\mathbb{Z}_N$ case without any difficulty. Furthermore, our discussion will primarily center on scenarios where both lattice sizes, $L_x$ and $L_y$, are even. To say it in advance, we have numerically confirmed that the PEPS achieved by the discussion below also serves as a MES for lattice size under various conditions.

In $\mathbb{Z}_2$ case, one can understand the virtual legs as being occupied by a loop when the corresponding virtual index has a value of 1, and being unoccupied when the the value is 0. Using this interpretation, we can translate the PEPS wavefunction in Eq. (D.1) into the loop-gas configuration picture. In doing so, we discover that the PEPS wavefunction is essentially an equal superposition of every possible closed-loop configurations, both on the solid and dotted lattices, respectively. An example of the closed-loop configuration is given in Fig. 6 (a).

This is attributed to the role played by $T_{udlr}$, which acts to ensure the closure of all loops. It is because, from the definition of $T_{udlr}$ in Eq. (D.2), whenever a loop enters to $T_{udlr}$, it must also exit. It is important to note that the closed-loop condition enforced by $T_{udlr}$ facilitates the presence of non-contractible loop configurations around the torus system.

Now, we consider how the WW operator $W_1$ acts on the PEPS wavefunction. To see that, we need to understand the action of a local Pauli-$X$ operator on the $g$ tensors. Referring to the definition in Eq. (D.2), one can observe that when $s_1 = 1$ ($s_2 = 1$), a solid (dotted) line crossing the $g^{s_1 s_2}_{udlr}$ tensor is occupied by a horizontal (vertical) loop, while it remains empty when $s_1 = 0$ ($s_2 = 0$). Additionally, for either the dotted or solid lines crossing the $g^{s_0}_{udlr}$, they are occupied by horizontal or vertical lines when $s_0 = 1$, and both lines are either empty or

(a) (b)

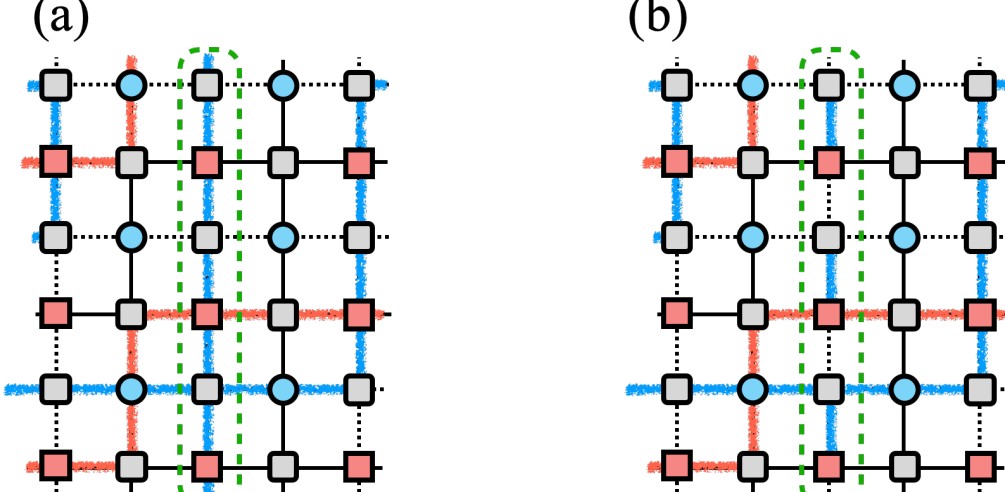

Figure 6: Illustrations of R2TC loop configurations. Blue lines denote loops on the dotted lattice, while orange lines represent loops on the solid lattice. The physical indices are omitted. The position where $W_6$ is applied is marked by closed green dashed-loop. (a) A configuration comprising exclusively of closed loops. (b) A configuration featuring a single isolated vertical loop. All other loops, apart from the vertical ones inside the green dashed rectangle, are closed.

occupied by both horizontal and vertical lines when $s_0 = 0$. Consequently, applying a local Pauli-$X$ operator to the $g$ tensors leads to a switch in the occupation state as described.

The WW operator $W_1$ acts by applying Pauli-$X$ operators to lattice sites positioned on the non-contractible dotted line defined along the $x$-axis. Building upon the insights from the previous paragraph, it becomes clear that the action of $W_1$ involves the insertion of a horizontal non-contractible loop in the dotted lattice. It is worth noting that in the context of our $\mathbb{Z}_2$ system, the insertion of a non-contractible loop is equivalent to toggling the number of non-contractible loops between even and odd. Since the PEPS wave function described in Eq. (D.1) already includes all possible closed loop configurations, encompassing both cases containing an even or odd number of non-contractible loops, the operations performed by the $W_1$ operator do not alter the PEPS wave function. This property results in the PEPS wavefunction being an eigenstate of $W_1$ with eigenvalue 1. By applying a similar rationale as discussed above, one can ascertain that the remaining WW operators, $W_2$, $W_3$, and $W_4$ also function by inserting horizontal or vertical non-contractible loops into the dotted or solid lattices, respectively. Employing analogous logic, it can be deduced that the PEPS wavefunction defined in Eq. (D.1) is likewise an eigenstate of $W_2$, $W_3$, and $W_4$ with eigenvalue 1.

Now, we will delve into the two steps involved in transforming the PEPS wavefunction into an MES within the framework of loop-gas configurations, addressing each step individually. The first step is to shift the PEPS wavefunction from being an eigenstate of $W_1$ and $W_3$ by applying the appropriate projection. Let's first consider the $W_1$ operator, with the same approach extendable to the $W_3$ operator. In the context of the loop-gas interpretation, the PEPS wavefunction can be expressed as $|0\rangle + |1\rangle$. Here, $|0\rangle$ represents an equal superposition of all loop configurations with an even number of loops crossing the entanglement $y$-cut, whereas $|1\rangle$ represents an equal superposition of all loop configurations with an odd number of loops crossing the entanglement $y$-cut. Each of them is a ground state of the R2TC. As described in the previous subsection, the action of $W_1$ toggles $|0\rangle$ into $|1\rangle$ and vice versa.

To satisfy the MES condition, one simply needs to project the PEPS wavefunction into either $|0\rangle$ or $|1\rangle$. In the loop-gas configuration picture, this translates to ensuring that the PEPS wavefunction contains either an even or odd number of horizontal non-contractible loops in the dotted lattice.

Expanding on this idea, one can observe that enforcing the PEPS wavefunction to be an eigenstate of $\widetilde{W_3}$ is equivalent to guaranteeing that the PEPS wavefunction comprises only even or odd numbers of horizontal non-contractible loops in the solid lattice. This marks the first step in the process of making the PEPS wavefunction an MES.

The final step involves ensuring that the PEPS wavefunction becomes an eigenstate of $W_6$. The original PEPS wavefunction presented in Eq. (D.1) consists exclusively of closed-loop configurations, as shown in Fig. 6 (a). However, applying $W_6$ introduces open-loop segments along the line where the WW operator is applied. For instance, when $W_6$ is applied to the vertical line marked by the dashed green rectangle in Fig. 6 (a), it affects every other edge, either breaking an existing closed loop or creating an open loop, as illustrated in Fig. 6 (b).

Let $|\psi_0\rangle$ denote the original PEPS wavefunction and $|\psi_1\rangle = W_6|\psi_0\rangle$ represent the wavefunction with open-loop segments resulting from the action of the WW operator. The action of $W_6$ toggles between $|\psi_0\rangle$ and $|\psi_1\rangle$. Since the MES must be an eigenstate of $W_6$, it should be of the form $|\psi_{\mathrm{MES}}\rangle = |\psi_0\rangle \pm |\psi_1\rangle$. This completes the final step of transforming the original PEPS wavefunction into an MES.

### D.3 Tensor network implementation

In the following, we will provide a detailed explanation of how to implement these two steps within the framework of TN one by one.

In the first step, we start by ensuring that the PEPS wavefunction contains only an even number of horizontal non-contractible closed loops on the dotted lattice. This is achieved by selecting any single vertical line from the dotted lattice and attaching a new tensor $\Omega_{udlr}$ to the $T_{udlr}$ tensors located along the vertical line as shown below:

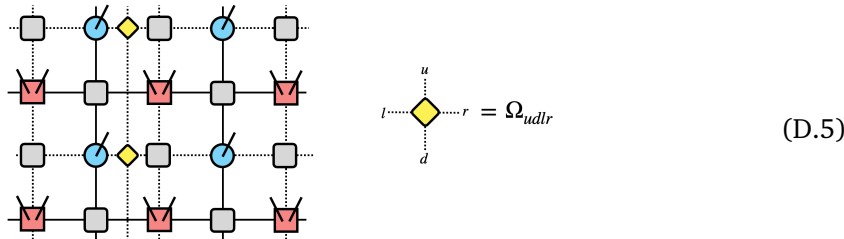

$$\tag{D.5}$$

The tensor $\Omega_{udlr}$ is defined as

$$\Omega_{udlr} = \delta_{l,r}\delta_{l,u-d}. \tag{D.6}$$

To understand how the tensor $\Omega_{udlr}$ constrains the number of horizontal non-contractible closed loops, let's focus on a vertical dotted bond connected to $\Omega_{udlr}$. Assume we start with a value of 0 for the virtual bond and move along the $y$-direction. Every time a horizontal loop crosses through $\Omega_{udlr}$, the value of the vertical bond toggles between 0 and 1, and vice versa. If a configuration has an odd number of non-contractible horizontal loops on the dotted lattice, we will encounter the horizontal loop an odd number of times during the round trip, leaving the virtual bond with a value of 1. Since the initial value of the virtual bond was 0, this leads to a mismatch, excluding the configuration from the tensor contraction. Thus, we obtain configurations consisting solely of an even number of horizontal non-contractible loops on the dotted lattice.

Similarly, by attaching an additional $\Omega_{udlr}$ tensor to the $T_{udlr}$ tensors located along any single vertical line from the solid lattice as follows:

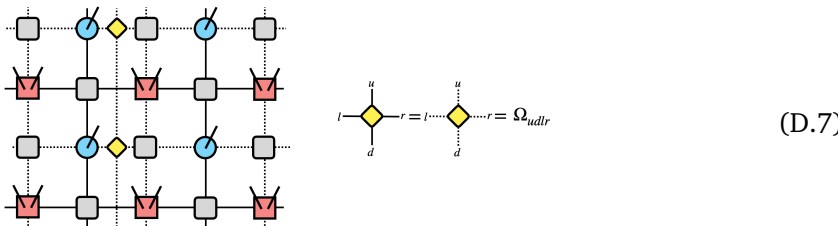

$$(D.7)$$

Note that $\Omega_{udlr}$ is not connected to the remaining vertical lattice lines. As a result, the PEPS wavefunction now exclusively comprises loop configurations with an even number of horizontal non-contractible loops on each lattice. This ensures that the PEPS wavefunction is shifted from being an eigenstate of $W_1$ and $W_3$, and the first step of transforming the PEPS wavefunction into an MES is successfully implemented.

The second step in ensuring the PEPS wavefunction becomes an MES involves more than just attaching additional auxiliary tensors. It also requires modifying certain local tensors $T_{udlr}$. Similar to the first step, we select a single vertical line from the dotted lattice and replace the $T_{udlr}$ tensors with $T_{udlr;m}$ tensors, which can be graphically illustrated as follows:

$$T_{udlr;m} \quad = \qquad \qquad = \qquad \qquad \qquad (D.8)$$

Here, the first image provides a top view, while the second shows a side view from a slanted perspective. The leg representing the $\mathbb{Z}_2$ index $m$ extends in the direction penetrating the surface, whereas the physical indices in Eq. (D.1) emerge from the surface. Its mathematical definition is given by

$$T_{udlr;m} = \delta_{u+r-(l+d),m}. \qquad \qquad (D.9)$$

Now, $T_{udlr;m}$ allows termination of a loop at the point when $m = 1$. Note that $T_{udlr;0} = T_{udlr}$.

Afterward, we attach the auxiliary tensor $\Theta_{ud}^m$ right below the $T_{udlr;m}$ as

$$\sum_m T_{udlr;m} \Theta_{u'd'}^m \quad = \qquad \qquad = \qquad \qquad \qquad (D.10)$$

Here, $\Theta_{u'd'}^m$ is defined as

$$\Theta_{u'd'}^m = \delta_{m,u'} \delta_{u,d'}. \qquad \qquad (D.11)$$

Consequently, the ultimate structure of the PEPS wavefunction takes the form:

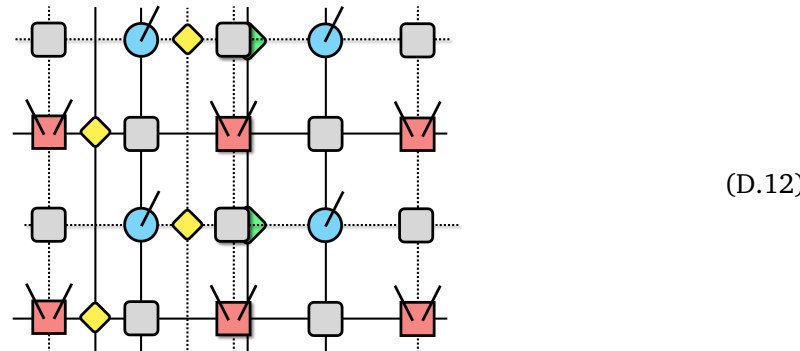

(D.12)

With the introduction of $\Theta^m_{u'd'}$, an additional vertical line is created on the "lower level" of the original lattice, linking the indices $u'$ and $d'$. As defined in Eq. (D.11), this vertical line allows only two configurations: either a closed non-contractible loop occupies the line, or the line remains empty. The configurations of the empty line correspond to the configurations prior to the second step of transforming the PEPS wavefunction into an MES. In the case of an occupied line, the configurations include a single vertical dashed loop on the dotted lattice, as depicted in Fig. 6 (b). Consequently, the resulting wavefunction is an eigenstate of $W_6$ with eigenvalue 1, completing the entire process of transforming the PEPS wavefunction into an MES.

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
