# Peer review of "Unveiling UV/IR Mixing via Symmetry Defects: A View from Topological Entanglement Entropy"

_SciPost Physics, doi:SciPost Phys. 18, 110 (2025)_

## Round 3 · Referee Report · Anonymous (Referee 1) · 2025-2-2

Report

In this paper, the authors calculated the topological entanglement entropy of Z_N rank-2 toric code model on a torus geometry. They considered both contractible boundary case (the boundary forms trivial loop) and noncontractible case (the boundary winds around the torus). The results are interesting and possibly generalized to more general boundaries like curved non-contractible boundaries, or other subdimensional particle phases.

I recommend publication of the paper. However, there is a minor point I want the authors to reconsider: - Just before (3.11), it is stated that the contribution to |G_B| from the third operator in (3.10) is N/gcd(L_x,N). However, it seems to be gcd(L_x,N) because there is an exponent c_x in the third operator.

Recommendation

Publish (meets expectations and criteria for this Journal)

---

## Round 3 · Referee Report · Anonymous (Referee 2) · 2025-2-4

Strengths

1- All the claims are very clearly argued for, with very explicit calculations 2 - Clearly written

Report

The paper studies in detail the rank-2 toric code model, from the perspective of symmetry defects. The authors argue that these can explain both the ground state degeneracy, as well as lattice size dependent entanglement entropy along non-contractible cuts. More specifically, the authors show, through explicit calculations, that the topological entanglement entropy (TEE) of the model's low energy states agree with the formula for TEE in SETs (arXiv:1410.4540).

The paper is well-written and meets all the journal's criteria. I have very few follow-up questions, but I believe they might enhance the overall quality of the paper and help clarify certain points

    - As it is an important topic to the paper, I think the introduction section should contain a literature review of translation SETs. From the top of my head, the connection between spatially modulated theories (which R2TC is an example of) and SETs enriched by spatial symmetries were explored in previous papers, e.g., arXiv:2204.07111 and 2310.09490.

    -  In Sec. IV the authors claim that the low energy physics of the R2TC can be thought of as a ZN x ZN translation SET. The anyons, however,  also transform non-trivially under rotations (e.g. mx and my, under pi/4 rotations). Could this be an  evidence this corresponds to some kind of rotation SET?

    -Still related to the previous question, can translations alone account for entanglement entropy related to diagonal non-contractible boundaries? Or one need rotations for that?

Requested changes

Can one address the questions raised above?

Recommendation

Publish (meets expectations and criteria for this Journal)

  • validity: good
  • significance: good
  • originality: high
  • clarity: top
  • formatting: excellent
  • grammar: excellent

Author:  Jintae Kim  on 2025-02-14  [id 5221]

(in reply to Report 2 on 2025-02-04)
Category:
answer to question

Comment: As it is an important topic to the paper, I think the introduction section should contain a literature review of translation SETs. From the top of my head, the connection between spatially modulated theories (which R2TC is an example of) and SETs enriched by spatial symmetries were explored in previous papers, e.g., arXiv:2204.07111 and 2310.09490.

Reply: We cite these papers in the explanation of translation SET and add a sentence to clarify the meaning of translation SET and its connection to spatially modulated gauge theories.

Comment: In Sec. IV the authors claim that the low energy physics of the R2TC can be thought of as a ZN x ZN translation SET. The anyons, however, also transform non-trivially under rotations (e.g. mx and my, under pi/4 rotations). Could this be an evidence this corresponds to some kind of rotation SET?

Reply: This is indeed an insightful comment. To clarify the referee's point further, a $\pi/2$ rotation about the reference point $(i_x, i_y) = (0,0)$, combined with the exchange of qudits at the same vertices (SWAP), results in the following transformations (see Eq.(2.2) for derivation):

$[e]_{i_x,i_y}^{l_1} \rightarrow [e]_{-i_y,i_x}^{l_1},$
$[m^x]_{i_x,i_y}^{l_2}\rightarrow[m^y]_{-i_y,i_x}^{l_2}$
$[m^y]_{i_x,i_y}^{l_3}\rightarrow[m^x]_{-i_y,i_x}^{l_3}.$

This implies that the R2TC exhibits a rotation+SWAP symmetry-enriched topological order, although this aspect lies beyond the scope of the present work. We mention this perspective in the Discussion.

Comment: Still related to the previous question, can translations alone account for entanglement entropy related to diagonal non-contractible boundaries? Or one need rotations for that?

Reply: We think considering the entanglement entropy and topological entanglement entropy for regions with diagonal non-contractible boundaries is a totally different problem which may not be associated with translation or rotation. The key challenge lies in the fact that, for an x- or y-cut, the maximal TEE is expected to correspond to an MES, which is a state that is a simultaneous eigenstate of all logical operators running along the cut. However, for diagonal non-contractible boundaries, the conventional definition of the MES is not directly applicable. Therefore, it is not clear whether the TEE for diagonal non-contractible boundaries is associated with some anyonic information.

On the other hand, one may inquire whether, when considering the creation and annihilation of rotation+SWAP symmetry defects along the x- or y-axis, and if the exact calculation of TEE for an x- or y-cut is feasible, Eq. (4.11) remains valid. This point is mentioned in the Discussion.

---

## Round 4 · Author Response

Warnings issued while processing user-supplied markup:

  • Inconsistency: plain/Markdown and reStructuredText syntaxes are mixed. Markdown will be used.
    Add "#coerce:reST" or "#coerce:plain" as the first line of your text to force reStructuredText or no markup.
    You may also contact the helpdesk if the formatting is incorrect and you are unable to edit your text.

Dear editors,

We thank you for sending us the report from referees on our manuscript, “UV/IR Mixing Via Symmetry Defects: A View from Topological Entanglement Entropy”. We have revised the manuscript to faithfully address the referees’ comments. Please see also our responses to various portions of the referee comments.

We would like to resubmit the revised manuscript along with the response letter.

yours sincerely, Jintae Kim, Yun-Tak Oh, Daniel Bulmash, and Jung Hoon Han

Report of Referee #1

Comment: Just before (3.11), it is stated that the contribution to |G_B| from the third operator in (3.10) is N/gcd(L_x,N). However, it seems to be gcd(L_x,N) because there is an exponent c_x in the third operator.

Reply: Thank you for noting the typo. We have revised it.

Report of Referee #2

Comment: As it is an important topic to the paper, I think the introduction section should contain a literature review of translation SETs. From the top of my head, the connection between spatially modulated theories (which R2TC is an example of) and SETs enriched by spatial symmetries were explored in previous papers, e.g., arXiv:2204.07111 and 2310.09490.

Reply: We cite these papers in the explanation of translation SET and add a sentence to clarify the meaning of translation SET and its connection to spatially modulated gauge theories.

Comment: In Sec. IV the authors claim that the low energy physics of the R2TC can be thought of as a ZN x ZN translation SET. The anyons, however, also transform non-trivially under rotations (e.g. mx and my, under pi/4 rotations). Could this be an evidence this corresponds to some kind of rotation SET?

Reply: This is indeed an insightful comment. To clarify the referee's point further, a $\pi/2$ rotation about the reference point $(i_x, i_y) = (0,0)$, combined with the exchange of qudits at the same vertices (SWAP), results in the following transformations (see Eq.(2.2) for derivation):

[e]{i_x,i_y}^{l_1} ->[e], [m^x]}^{l_1{i_x,i_y}^{l_2}->[m^y] [m^y]}^{l_2{i_x,i_y}^{l_3}->[m^x].}^{l_3

This implies that the R2TC exhibits a rotation+SWAP symmetry-enriched topological order, although this aspect lies beyond the scope of the present work. We mention this perspective in the Discussion.

Comment: Still related to the previous question, can translations alone account for entanglement entropy related to diagonal non-contractible boundaries? Or one need rotations for that?

Reply: We think considering the entanglement entropy and topological entanglement entropy for regions with diagonal non-contractible boundaries is a totally different problem which may not be associated with translation or rotation. The key challenge lies in the fact that, for an x- or y-cut, the maximal TEE is expected to correspond to an MES, which is a state that is a simultaneous eigenstate of all logical operators running along the cut. However, for diagonal non-contractible boundaries, the conventional definition of the MES is not directly applicable. Therefore, it is not clear whether the TEE for diagonal non-contractible boundaries is associated with some anyonic information.

On the other hand, one may inquire whether, when considering the creation and annihilation of rotation+SWAP symmetry defects along the x- or y-axis, and if the exact calculation of TEE for an x- or y-cut is feasible, Eq. (4.11) remains valid. This point is mentioned in the Discussion.

---

## Round 4 · List of Changes

1. Revised the typo
  2. Revised the Discussion.

---

## Editorial Decision

published